

# Controls on spatial and temporal variability of streamflow
# and hydrochemistry in a glacierized catchment
**Running title: Controls on streamflow and hydrochemistry in a glacierized catchment**
Michael Engel[1], Daniele Penna[2], Giacomo Bertoldi[3], Gianluca Vignoli[4], Werner Tirler[5,] and
Francesco Comiti[1]
[1]Faculty of Science and Technology, Free University of Bozen-Bolzano, Piazza Università 5,
39100 Bozen-Bolzano, Italy
[2]Department of Agricultural, Food and Forestry Systems, Via S. Bonaventura, 13, University
of Florence, 50145 Florence, Italy
[3]Institute for Alpine Environment, Eurac Research, Viale Druso 1, 39100 Bozen-Bolzano,
Italy
[4]CISMA S.r.l., Via Volta 13/A, 39100 Bozen-Bolzano, Italy
[5]Eco-Research S.r.l., Via Negrelli 13, 39100 Bozen-Bolzano, Italy
*Correspondence to*: Michael Engel (Michael.Engel@unibz.it)
**Abstract**
The understanding of the hydrological and hydrochemical functioning of glacierized
catchment requires the knowledge of the different controlling factors and their mutual
interplay. For this purpose, the present study was carried out in two sub-catchments of the
Sulden River catchment (130 km², Eastern Italian Alps) in 2014 and 2015, characterized by
similar size but contrasting geological setting. Samples were taken at different space and time
scales for analysis of stable isotopes of water, electrical conductivity, major, minor and trace
elements.
At the monthly sampling scale for different spatial scales (0.05 – 130 km²), complex spatial
and temporal dynamics such as contrasting EC gradients in both sub-catchments were found.
At the daily scale, for the entire Sulden catchment the relationship between discharge and
electrical conductivity showed a monthly hysteretic pattern. Hydrometric and geochemical
dynamics were controlled by an interplay of meteorological conditions and geological
heterogeneity. After conducting a PCA analysis, the largest share of variance (36.3 %) was





explained by heavy metal concentrations (such as Al, V, Cr, Ni, Zn, Cd, Pb) during the
melting period while the remaining variance (16.3 %) resulted from the bedrock type in the
upper Sulden sub-catchment (inferred from EC, Ca, K, As and Sr concentrations). Thus, high
concentrations of As and Sr in rock glacier outflow may more likely result from bedrock
weathering. Furthermore, nivo-meteorological indicators such as maximum daily global solar
radiation, three day maximum air temperature, and 15 day snow depth differences could
explain the monthly conductivity and isotopic dynamics best. The decrease of snow depth
calculated for different time lengths prior to the sampling day showed best agreements with
conductivity and isotopic dynamics when time lengths varied. These insights may help to
better predict hydrochemical catchment responses linked to meteorological and geological
controls and to guide future classifications of glacierized catchments according to their
hydrochemical characteristics.

## 1    Introduction

Runoff from glacierized catchments is an important fresh water resource to downstream areas
(Kaser et al., 2010; Viviroli et al., 2011). High-elevation environments face rapid and
extensive changes through retreating glaciers, reduced snow cover, and permafrost thawing
(Harris et al., 2001; Dye, 2002; Beniston, 2003; Galos et al., 2015). This will have impacts on
runoff seasonality, water quantity and water quality (Beniston 2006; Ragettli et al., 2016;
Gruber et al., 2017). It is therefore of uttermost importance to better understand the behaviour
of high-elevation catchments and their hydrological and hydrochemical responses at different
spatial and temporal scales in view of water management, water quality, hydropower, and
ecosystem services under the current phase of climate change (Beniston, 2003; Viviroli et al.,
2011; Beniston and Stoffel, 2014).
In general, the hydrological response of catchments (i.e. runoff dynamics) are controlled by
heterogeneous catchment properties (Kirchner, 2009), which become more diverse in
catchments with large complexity of various landscape features, as it is the case of
mountainous, high-elevation glacierized catchments (Cook and Swift, 2012). In fact, those
catchments    are    deemed    as    highly    dynamic    geomorphological,    hydrological    and
biogeochemical environments (Rutter et al., 2011). Understanding the interactions of controls
driving the catchment response represents the key focus of studies in catchment hydrology



(Troch et al., 2015). The advances of tracer and isotope hydrology made during the last
decades can substantially contribute to this objective, in order to gain more insights into the
variability of different runoff components (Vaughn and Fountain, 2005; Maurya et al., 2011;
Xing et al., 2015), catchment conceptualization (Baraer et al., 2015; Penna et al., 2017), and
sensitivity to climate change (Kong and Pang, 2012).
In general, the main controls of hydrological and hydrochemical catchment responses are
represented by climate, bedrock geology, surficial geology, soil, vegetation, and topography
with drainage network (Devito et al., 2005; Carrillo et al., 2011; Williams et al 2015) and
catchment shape (Sivapalan 2003). First, a major role is attributed to the global and regional
climate, having strong impacts on mountain glaciers and permafrost, streamflow, water
quality, water temperature, and suspended sediment yield (Milner et al., 2009; Moore et al.,
2009; IPCC, 2013). The impact of climate is difficult to assess because it requires long time
windows (e.g., decades), whereas meteorological drivers interact at a smaller temporal scales
and thus are easier to address. Among different meteorological drivers, radiation fluxes at the
daily time scale were identified as main energy source driving melting processes in
glacierized catchments in different climates (Sicart et al., 2008). Beside radiation, air
temperature variations correlate well with runoff under the presence of snow cover (Swift et
al., 2005) and may affect streamflow seasonality when specific thresholds are exceeded
(Cortés et al., 2011).
With respect to geology, it sets the initial conditions for catchment properties and drives its
evolution (Carrillo et al., 2011). The geological setting strongly controls catchment
connectivity, drainage, and groundwater discharge (Farvolden 1963), runoff response (Onda
et al., 2001), residence time (Katsuyama et al., 2010), hydrochemistry during baseflow
conditions (Soulsby et al., 2006a) and melting periods (Hindshaw et al., 2011), and subglacial
weathering (Brown and Fuge, 1998). Also geomorphological features such as talus fields may
affect streamflow and water quality, resulting from different flow sources and flow pathways
(Liu et al., 2004). Catchment storage, as determined by both geology and topography, was
found to impact the stream hydrochemistry as well (Rinaldo et al., 2015).
The hydrological conditions of the catchment are also a relevant driver of hydrological
response and commonly refer to the antecedent soil moisture conditions to describe the state
of the catchment and represent the hydrological connectivity (Uhlenbrook and Hoeg, 2003;
Freyberg et al., 2017). Specifically in high elevation and high latitude catchments, also



permafrost thawing affects the hydrological connectivity (Rogger et al., 2017), leading to a strong control on catchment functioning as it drives the partitioning, storage and release of water (Tetzlaff et al., 2014). In more detail, retreating permafrost may also result in distinct geochemical signatures (Clark et al., 2001) and the release of heavy metals being previously stored in the ice (Thies et al., 2007; Krainer et al., 2015). This does not affect only the water quality but also the aquatic biota such as macroinvertebrate communities in these environments (Milner et al., 2009). Different weathering processes between the subglacial and periglacial environment can be found, resulting in a shift in chemical species and concentrations in the water (Anderson et al., 1997).

However, only few studies have investigated the geological, meteorological, and topographic controls on catchment response and stream water hydrochemistry in glacierized or permafrost-dominated catchments (Wolfe and English, 1995; Hodgkins, 2001; Lewis et al., 2012).

In this paper, we aim to fill this gap presenting data from a two year monitoring campaign where samples for stable isotopes of water, electrical conductivity (EC), major, minor and trace elements analysis were collected for two nearby glacierized catchments in the Eastern Italian Alps, characterized by similar size and climate and but contrasting geological setting.

The present study builds up on the following hypotheses: (1) bedrock-specific geochemical signatures reveal the geographic origin of water sources, (2) dilution effects and isotopic depletion in stream hydrochemistry are explained better by nivo-meteorological indicators controlling melt processes by radiation and air temperature than by precipitation-related indicators and (3) catchment controls not varying in short periods (such as geology and topography) lead to spatial variation in hydrochemistry while short-term controls (such as meteorological conditions) affect the temporal variations of hydrochemistry.

Specifically, we aim to:

- assess the spatio-temporal variability of the hydrochemical signature of stream water during melting and baseflow conditions;

- identify the hydrochemical signature of thawing permafrost and its role on stream water;



• analyse the capability of nivo-meteorological indicators to describe the
hydrochemical signature of stream water.
**2 Study area and instrumentation**
**2.1 The Sulden river catchment**
The study was carried out in the Sulden/Solda River catchment, located in the upper
Vinschgau/Venosta Valley (Eastern Italian Alps) (Fig. 1). The size of the study area is about
130 km² defined by the stream gauge station of the Sulden River at Stilfserbrücke/ Ponte
Stelvio (1110 m a.s.l.). The highest elevation is represented by the Ortler/ Ortles peak (3905
a.s.l.) within the Ortles-Cevedale group. A major tributary is the Trafoi River, joining the
Sulden River close to the village Trafoi-Gomagoi. At this location, two sub-catchments,
namely Sulden and Trafoi sub-catchment (75 and 51 km², respectively) meet.
The study area has a current glacier extent of about 17.7 km² (14 % of the study area) and is
slightly higher in the Trafoi than in the Sulden sub-catchment (17 % and 12 %, respectively).
Main glacier tongues in the study area are represented by the Madatsch glacier (Trafoi sub-
catchment) and Sulden glacier (Sulden sub-catchment). Geologically, the study area belongs
to the Ortler-Campo-Cristalin (Mair et al., 2007). While permotriassic sedimentary rocks
dominate the Trafoi sub-catchment, Quarzphyllite, Orthogneis, and Amphibolit are present in
the Sulden sub-catchment. However, both catchments share the presence of orthogneis,
paragneis and mica schist from the lower reaches to the outlet. Permafrost is sparsely located
between 2400 and 2600 m a.s.l. and more frequent above 2600 m a.s.l. (Boeckli et al., 2012).
Climatically, the mean annual air temperature is about -1.6 °C and the mean annual
precipitation is about 1008 mm (2009 - 2016) at 2825 m a.s.l. (Hydrographic Office,
Autonomous Province of Bozen-Bolzano). Due to the location of the study area in the inner
dry Alpine zone, these precipitation amounts are relatively low compared to the amounts at
similar elevation in the Alps (Schwarb, 2000). Further climatic data regarding the sampling
period of this study are shown in Table 1.The study area lies within the National Park "Stelvio
/ Stilfser Joch" but it also includes ski slopes and infrastructures, as well as hydropower weirs.





## 2.2 Meteorological, hydrometric and topographical data

Precipitation, air temperature, humidity and snow depth is measured by an ultrasonic sensor at 10 min measuring interval at the automatic weather station (AWS) Madritsch/Madriccio at 2825 m a.s.l. (run by the Hydrographic Office, Autonomous Province of Bozen-Bolzano). We take data from this station as representative for the glacier in the catchment at similar elevation. At the outlet at Stilfserbrücke/Ponte Stelvio, water stages are continuously measured by an ultrasonic sensor (Hach Lange GmbH, Germany) at 10 min measuring interval and converted to discharge via salt dilution/photometric measurements (measurement range: $1.2 - 23.2$ m³ s$^{-1}$; n=22). Turbidity is measured by a SC200 turbidity sensor (Hach Lange GmbH, Germany) at 5 min measuring interval. EC is measured by a TetraCon 700 IQ (WTW GmbH, Germany) at 1 second measuring interval. Both datasets were resampled to 10 min time steps. All data used in this study are recorded and presented in solar time.

Topographical data (such as catchment area and 50 m elevation bands) were derived from a 2.5 m DEM using GIS processing (ArcGIS 10, ESRI).

## 2.3 Tracer sampling and analysis

Continuous stream water sampling at the outlet was performed by an automatic sampling approach using an ISCO 6712 system (Teledyne Technologies, USA). Generally, daily water sampling took place from mid-May to mid-October 2014 and 2015 (on 331 days) at 23:00 to ensure consistent water sampling close to the discharge peak and respecting its seasonal variation. In addition, grab samples from different stream locations, tributaries, and springs in the Sulden and Trafoi sub-catchments and the outlet were taken monthly from February 2014 to November 2015 (Table 2). Samples were collected approximately at the same time (within less of an hour of difference) on all occasions. In winter, however, a different sampling time had to be chosen for logistical constraints (up to four hours of difference between both sampling times).However, this did not produce a bias on the results due to the very limited variability of the hydrochemical signature of water sources during winter baseflow conditions. Two active rock glaciers, located on Quarzphyllite bedrock in the upper Sulden sub-catchment, were selected to represent meltwater from permafrost. At the base of the steep rock glacier front, three springs at about 2600 m a.s.l. were sampled monthly from July to September 2014 and July to October 2015. Snowmelt water was collected as dripping water



from snow patches from April to September 2014 and March to October 2015 (n = 48
samples), mainly located on the west to north-facing slopes of the Sulden sub-catchment and
at the head of the valley in the Trafoi sub-catchment. Glacier melt water was taken only at the
eastern tongue of the Sulden glacier from July to October 2014 and 2015 (n = 11 samples) for
its safe accessibility. Precipitation samples were derived from bulk precipitation collectors,
built according to the standards of the International Atomic Energy Agency (International
Atomic Energy Agency 2014). They were placed at four different locations covering an
elevations gradient of 1750 m and emptied on a monthly basis from April to November 2014
and 2015. Only the precipitation collector at the mountain hut Schaubach remained during
winter 2014/2015 to collect winter precipitation. Due to limited accessibility mainly in spring
and autumn, the collector was emptied after more than one month. Snow samples were
derived from snow profiles as integrated and layer-specific samples, which were dug along an
elevation gradient once a month from January to April 2015 and after snowfall events in
August to October 2015.
EC was measured in the field by a portable conductivity meter WTW 3410 (WTW GmbH,
Germany) with a precision of +/- 0.1 $\mu$S cm$^{-1}$ (nonlinearly corrected by temperature
compensation at 25 °C).
All samples were stored in 50 ml PVC bottles with a double cap and no headspace. The
samples were kept in the dark at 4°C in the fridge before the analysis. $\delta^2$H and $\delta^{18}$O isotopic
composition of all water samples (except the ISCO stream water samples at the outlet) were
analysed at the Laboratory of Isotope and Forest Hydrology of the University of Padova
(Italy), Department of Land, Environments, Agriculture and Forestry by an off-axis integrated
cavity output spectroscope (model DLT-100 908-0008, Los Gatos Research Inc., USA). The
analysis protocol and the description of reducing the carry-over effect are reported in (Penna
et al., 2010, 2012). The instrumental precision (as an average standard deviation of 2094
samples) is 0.5‰ for $\delta^2$H and 0.08‰ for $\delta^{18}$O.
The $\delta^{18}$O isotopic composition of the ISCO stream water samples was analysed by an isotopic
ratio mass spectrometer (GasBenchDelta V, Thermo Fisher) at the Free University of Bozen-
Bolzano. Following the gas equilibration method (Epstein and Mayeda, 1953), 200-$\mu$l sub-
samples were equilibrated with He–$CO_2$ gas at 23 °C for 18 h and then injected into the
analyser. The isotopic composition of each sample was calculated from two repetitions, and
the standard deviation was computed. The instrumental precision for $\delta^{18}$O was ±0.2‰. We

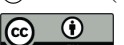



applied a correction factor, described in Engel et al. (2016), to adjust the isotopic
compositions of $\delta^{18}O$ measured by the mass spectrometer to the ones measured by the laser
spectroscope.
The analysis of major, minor and trace elements (Li, B, Na, Mg, Al, K, Ca, V, Cr, Mn, Fe,
Co, Ni, Cu, Zn, Rb, Sr, Mo, Ba, Pb and U) was carried out by Inductively Coupled Plasma
Mass Spectroscopy (ICP-MS ICAP-Q, Thermo Fischer) at the laboratory of EcoResearch srl.
(Bozen-Bolzano).

### 2.4 Data analysis

In order to better understand the effect of meteorological controls at different time scales, in
particular precipitation and melting rates, different environmental variables derived from
precipitation, air temperature, solar radiation and snow depth data from AWS Madritsch, were
calculated (Table 3). Then, a sensitivity analysis was performed, which was based on a 1 day
incremental time step and a temporal length of 30 days to respect the period of time between
the monthly stream water samplings. As precipitation indicators, we considered the cumulated
precipitation P in a period between 1 and 30 days prior to the sampling day, and the period of
time $D_{prec}$ in days starting from 1, 10 or 20 mm of cumulated precipitation occurred prior to
the sampling day. As snow and ice melt indicators, we selected the maximum air temperature
$T_{max}$ and maximum global solar radiation $G_{max}$ in a period between 1 and 30 days prior the
sampling day. Moreover, we calculated the difference of snow depth $\Delta SD$ measured at the
sampling day and the previous days, varying from 1 to 30 days. The temporal sensitivities of
agreement between nivo-meteorological indicators and tracer signatures were expressed as
Pearson correlation coefficients ($p < 0.5$) and represented a measure to obtain the most
relevant nivo-meteorological indicators.
In order to understand the link among water sources and their hydrochemical composition, a
principle component analysis (PCA), using data centred to null and scaled to variance one (R
core team, 2016), was performed. Data below detection limit were excluded from the
analysis.
To assess the dampening effect of meltwater on stream water chemistry during baseflow
conditions and the melting period, the variability coefficient (VC) was calculated following
Eq. (1):
Variability coefficient $VC = SD_{baseflow}/SD_{melting}$          (1)

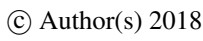


$SD_{baseflow}$ is the standard deviation of stream EC sampled during baseflow conditions in winter
at a given location and $SD_{melting}$ is the one at the same locations during the melt period in
summer (following Sprenger et al., 2016).
A two-component hydrograph separation (HS) based on EC and $\delta^2H$ was assigned to separate
the runoff contributions originating from the Sulden and Trafoi sub-catchment at each
sampling moment during monthly sampling (Sklash and Farvolden, 1979), following Eq. (2)
and Eq. (3):
$Q_{S1} = Q_{S2} + Q_{T1}$ (2)
$P_{T1} = (C_{S2} - C_{S1})/(C_{S2} - C_{T1})$ (3)
where P is the runoff proportion, C is the electrical conductivity EC or isotopic composition
in $^2H$ measured at the locations S1 (outlet), S2 (sampling location in the Sulden sub-
catchment upstream the confluence with Trafoi River), and T1 (sampling location in the
Trafoi sub-catchment upstream the confluence with Sulden River). While T1 served as "old
water" component, S2 represented the "new water" component at S1. The uncertainty in the
two-component HS was expressed as Gaussian error propagation using the instrumental
precision of the conductivity meter (0.1 $\mu S\ cm^{-1}$) and sample standard deviation from the laser
spectroscope, following Genereux (1998). Furthermore, statistical analysis were performed to
test the variance of hydrochemical data by means of a t-test (if data followed normal
distribution), otherwise the nonparametric Mann-Whitney test was used.
**3   Results**
**3.1   Origin of water sources**
The isotopic signature of all water samples collected in the study area is shown in Fig. 2.
Based on the isotopic signature of precipitation samples, the Local Meteoric Water Line
(LMWL) was close to the Global Meteoric Water Line (GMWL). The isotopic signature of
the other water sources fell on the water line, indicating that they originated from the same
water vapour source as precipitation, with no or negligible secondary post-depositional
fractionation. In more detail, rainfall samples represented the most enriched water source in
the catchment ($\delta^2H$: -128.6 to -15.14 ‰) while snow was the most depleted one ($\delta^2H$: -196.3
to -86.7 ‰) and became more enriched through melting processes, with a smaller isotopic
variability ($\delta^2H$: -137.33 to -88.0 ‰). In contrast, glacier melt and rock glacier spring water





were isotopically relatively similar and slightly more positive than snowmelt ($\delta^2$H: -105.7 to -
82.2 ‰, and -113.9 to -90.6 ‰, respectively). The isotopic range of spring water from the
valley bottom (TSPR1-2, SSPR1) was relatively similar to the one of snowmelt ($\delta^2$H: -105.7
to -88.8 ‰), with slightly more enriched samples from the Trafoi sub-catchment than from
the Sulden sub-catchment. Only few water samples (i.e. snowmelt samples) plotted below the
LMWL likely as a result of kinetic, non-equilibrium isotopic fractionation during the
snowpack melting process (inset of Fig. 2).
To identify the geographic origin of stream water within the catchment, element
concentrations of stream and rock glacier spring water are presented in Table 4 and 5. It is
worth highlighting that heavy metal concentrations (such as Al, V, Cr, Ni, Zn, Cd, Pb)
showed highest concentrations during intense melting in July 2015 at all six locations (partly
exceeding concentration thresholds for drinking water (see European Union (Drinking Water)
Regulations 2014). Element concentrations were clearly higher at the most upstream sampling
locations. Relatively low variability coefficients (VC < 0.3) for these elements confirmed that
larger variations of concentrations occurred during the melting period and not during
baseflow conditions. Interestingly, the highest heavy metal concentrations (such as Mn, Fe,
Cu, Pb) of rock glaciers springs SPR2 – 4 delayed the heavy metal concentration peak in the
stream by about two months.
In contrast, other element concentrations (such as As, Sr, K, Sb) generally revealed higher
concentrations during baseflow conditions and lower concentrations during the melting
period. This observation was corroborated by relatively high variability coefficients for As
(VC: 2 – 2.9) and Sb (VC: 2 – 2.2) at S1, S2, and T1. For example, while highest Sr
concentrations were measured at S6, As was highest at the downstream locations T1, S2, and
S1. Regarding the rock glacier springs, their hydrochemistry showed a gradual decrease in As
and Sr concentration from July to September 2015. The observed geochemical patterns are
confirmed by PCA results (Fig. 3) and the correlation matrix (Fig. 4), revealing that
geochemical dynamics are driven by temporal (PC1) and spatial controls (PC2) and a typical
clustering of elements, respectively. PC1 shows high loadings for heavy metal concentrations
(such as Al, V, Cr, Ni, Zn, Cd, Pb), supporting the clear temporal dependency for the entire
catchment (baseflow conditions vs. melting period)(Fig. 3a). PC2 is instead mostly
characterized by high loadings of $\delta^2$H and $\delta^{18}$O in the Trafoi sub-catchment (i.e. T1 and TT2)
and geochemical characteristics (EC, Ca, K, As and Sr) from the upstream region of the





Sulden River and rock glacier spring water (i.e. S6 and SSPR2-4, respectively). Overall,
temporal and spatial controls explained a variance of about 53 %.

## 3.2  Temporal and spatial tracer variability

The temporal and spatial variability of EC in the Sulden and Trafoi River along the different
sections, their tributaries, and springs is illustrated in Fig. 5. Results highlight the dominant
impact of water enriched in solutes during baseflow conditions starting from late autumn to
early spring prior to the onset of the melting period. Such an impact seemed to be highest in
water from streams and tributaries reaching the most increased conductivity at S6, ranging
from 967 to 992 $\mu$S cm$^{-1}$ in January to March 2015. During the same period of time, isotopic
composition was slightly more enriched and spatially more homogeneous among the stream,
tributaries, and springs than in the summer months. In contrast, during the melting period,
water from all sites in both sub-catchments became diluted due to different inputs of
meltwater (Fig. 5 a, b), while water was most depleted during snowmelt dominated periods
and less depleted during glacier melt dominated periods (Fig. 5c and 5d). Rainfall became a
dominant runoff component during intense storm events. For instance, on 24 September 2015,
a storm of 35 mm d$^{-1}$ resulted in the strongest isotopic enrichment of this study, which is
visible in Fig. 5c at T3 and TT2 ($\delta^2$H -86.9 ‰; $\delta^{18}$O: -12.4 ‰).
Hereinafter, the hydrochemistry of the Sulden and Trafoi sub-catchment is analyzed in terms
of hydrochemical patterns of the main stream, tributaries, springs, and runoff contributions at
the most downstream sampling location above the confluence. At T1 and S2, hydrochemistry
was statistically different in its isotopic composition (Mann-Whitney Rank Sum Test: $p <$
0.001) but not in EC (Mann-Whitney Rank Sum Test: $p = 0.835$). Runoff originating from
Trafoi and derived from the two-component HS, contributed to the outlet by about 36 %
($\pm 0.004$) to 58 % ($\pm 0.003$) when using EC and ranged from 29 % ($\pm 0.09$) to 83 % ($\pm 0.15$)
when using $\delta^2$H. Thus, runoff at the outlet was sustained more strongly by the Trafoi River
during non-melting periods while the runoff from the Sulden sub-catchment dominated during
the melting period.
By the aid of both tracers, catchment specific hydrochemical characteristics such as
contrasting EC gradients along the stream were revealed (Fig. 5 and Fig. 6). EC in the Trafoi



River showed linearly increasing EC with increasing catchment area (from T3 to T1) during
baseflow and melting periods ('EC enrichment gradient').
In contrast, the Sulden River revealed relatively high EC at the highest upstream location (S6)
and relatively low EC upstream the confluence with the Trafoi River (S2) during baseflow
conditions. The exponential decrease in EC ('EC dilution gradient') during this period of time
was strongly linked to the catchment area. Surprisingly, the EC dilution along the Sulden
River was still persistent during melting periods but highly reduced. In this context, it is also
interesting to compare the EC variability (expressed as VC) along Trafoi and Sulden River
during baseflow conditions and melting periods (Table 6). For both streams, VC increased
with decreasing distance to the confluence (Trafoi River) and the outlet (Sulden River), and
thus representing an increase in catchment size. The highest EC variability among all stream
sampling locations is given by the lowest VC, which was calculated for S6. This location
represents the closest one to the glacier terminus and showed a pronounced contrast of EC
during baseflow conditions and melting periods (see Fig. 5 and Fig. 6).
Regarding the hydrochemical characterisation of the tributaries in both sub-catchments (Fig.
5), Sulden tributaries were characterised by a relatively low EC variability (68.2 – 192.3 μS
cm$^{-1}$) and more negative isotopic values (δ²H: -100.8 – 114.5 ‰) compared to the higher
variability in hydrochemistry of the Sulden River. In contrast, the tracer patterns of Trafoi
tributaries were generally consistent with the ones from the stream. Generally, also spring
water at TSPR1, TSPR2, and SSPR1 followed these patterns during baseflow and melting
periods in a less pronounced way, possibly highlighting the impact of infiltrating snowmelt
into the ground. Comparing both springs sampled in the Trafoi sub-catchment indicated that
spring waters were statistically different only when using EC (Mann-Whitney Rank Sum
Test: $p = 0.039$). While TSPR1 hydrochemistry was slightly more constant, the one of TSPR2
was more variable from June to August 2015 (Fig. 5). This may result from different flow
paths and disconnected recharge areas sustaining separately each spring, possibly pointing to
a deeper (for TSPR1) and a shallower (for TSPR2) groundwater body.
**3.3    Temporal variability at the catchment outlet**
The temporal variability of the hydrochemical variables observed at the catchment outlet and
of the meteorological drivers is illustrated in Fig. 7. Controlled by increasing radiation inputs
and air temperatures above about 5°C in early summer (Fig. 7a and 7b), first snowmelt (as



indicated by a depleted isotopic signature of about -14.6 ‰ in $\delta^{18}$O and EC of about 200 μS
cm$^{-1}$) induced runoff peaks in the Sulden River of about 20 m³ s$^{-1}$ (starting from a winter
baseflow of about 1.8 m$^3$ s$^{-1}$), as shown in Fig. 7c and 7e. Later in the summer, glacier melt
induced runoff peaks reached about 13 – 18 m³ s$^{-1}$, which are characterised by relatively low
EC (about 235 μS cm$^{-1}$) and isotopically more enriched stream water ($\delta^{18}$O: about -13.3 ‰).
The highest discharge measured during the analysed period (81 m³ s$^{-1}$ on 13 August 2014)
was caused by a storm event, characterized by about 31 mm of precipitation falling over 3
hours at AWS Madritsch. Unfortunately, isotopic data for this event were not available due to
a technical problem with the automatic sampler.
Water turbidity was highly variable at the outlet, and mirrored the discharge fluctuations
induced by meltwater or storm events. Winter low flows are characterised by very low
turbidity (< 10 NTU, corresponding to less than 6 mg l$^{-1}$). In summer, turbidity ranged
between 20 and up to 1200 NTU during cold spells and melt events combined with storms,
respectively. However, the maximum value recorded was 1904 NTU reached after several
storm events of different precipitation amounts (17 mm, 50 mm, and 9 mm) on 12, 13, and 14
August 2014, respectively. Unfortunately, the turbidimeter did not work properly after the
August 2014 flood peak, in mid-July 2015 and beginning of October 2015.
Furthermore, the interannual variability of meteorological conditions with respect to the
occurrence of warm days, storm events and snow cover of the contrasting years 2014 and
2015 is clearly visible and contributed to the hydrochemical dynamics (Fig.7 and Table 1).
While about 250 cm of maximal snowpack depth in 2014 lasted until mid-July, only about
100 cm were measured one year after with complete disappearance of snow one month
earlier. In 2015, several periods of remarkable warm days occurred reaching more than 15°C
at 2825 m a.s.l. and led to a catchment entirely under melting conditions (freezing level above
5000 m a.s.l., assuming a lapse rate of 6.5 K km$^{-1}$). In contrast, warmer days in 2014 were less
pronounced and frequent but accompanied by intense storms of up to 50 mm d$^{-1}$. These
meteorological conditions seem to contribute to the general hydrochemical patterns described
above. Despite a relatively similar hydrograph with same discharge magnitudes during melt-
induced runoff events in both years, EC and $\delta^{18}$O clearly characterized snowmelt and glacier
melt-induced runoff events in 2014. However, a characteristic period of depleted or enriched
isotopic signature was lacking in 2015 so that snowmelt and glacier melt-induced runoff
events were graphically more difficult to distinguish.



The daily variations in air temperature, discharge, turbidity, and EC showed marked
differences in the peak timing. Maximum daily air temperature generally occurred between
12:00 and 15:00, resulting in discharge peaks at about 22:00 to 1:00 in early summer and at
about 16:00 to 19:00 during late summer. Turbidity peaks were measured at 22:00 to 23:00 in
May to June and clearly anticipated to 16:00 to 19:00 in July and August. In contrast, EC
maximum occurred shortly after the discharge peak between 00:00 to 1:00 in early summer
and at 11:00 to 15:00, clearly anticipating the discharge peaks.
It is interesting to highlight a complex hydrochemical dynamics during the baseflow period in
November 2015, which was interrupted only by a rain-on-snow event on 28 and 29 October
2015. This events was characterized by more liquid (12.9 mm) than solid precipitation (6.6
mm) falling on a snowpack of about 10 cm (at 2825 m a.s.l.). While stream discharge showed
a typical receding hydrograph confirmed by EC being close to the background value of about
350 $\mu$S cm$^{-1}$, $\delta^{18}$O indicated a gradual isotopic depletion suggesting the occurrence of
depleted water (e.g., snowmelt) in the stream. Indeed, also turbidity was more variable and
slightly increased during this period.
To better characterize the temporal dynamics of hydrochemical variables, Fig. 8 shows the
different relationships of discharge, EC, $\delta^{18}$O, and turbidity grouped for different months. In
general, high turbidity seemed to be linearly correlated with discharge showing a monthly
trend (Fig. 8a). In fact, this observation could be explained by generally higher discharges
during melting periods (June, July, and August) and lower ones during baseflow conditions.
Discharge and EC exhibited a relationship characterised by a hysteretic-like pattern at the
monthly scale (Fig. 8b), which seemed to be associated with the monthly increasing
contribution of meltwater with lower EC during melting periods contrasting with dominant
groundwater contributions having higher EC during baseflow conditions.
During these periods, $\delta^{18}$O of stream water was mainly controlled by the dominant runoff
components (i.e. snowmelt and glacier melt in early summer and mid- to late summer,
respectively) rather than the amount of discharge (Fig. 8c). Similarly, the relationship
between $\delta^{18}$O and EC was driven by the discharge variability resulting in a specific range of
EC values for each month and by the meltwater component generally dominant during that
period (Fig. 8d). As $\delta^{18}$O was dependent on the dominant runoff components and less on the
amount of discharge, turbidity showed no clear relationship with the isotopic composition



(Fig. 8e). In contrast, EC and turbidity were controlled by monthly discharge variations so
that both variables followed the monthly trend, revealing a linear relationship (Fig. 8f).
**3.4    Meteorological controls on hydrochemical stream responses within the catchment**
To identify the most significant correlations between stream hydrochemistry ($\delta^2$H and EC)
and nivo-meteorological indicators (Table 3), the Pearson correlation coefficient was used.
While significant correlations were generally found for maximum air temperature $T_{max}$ (only
for EC), maximum global solar radiation $G_{max}$, and the difference of snow depth $\Delta$SD, other
indicators such as cumulated precipitation $P_{cum}$ and $D_{Prec}$ were not significant ($p < 0.05$) and
thus excluded from further analysis.
As the correlation of the most relevant nivo-meteorological indicators $T_{max}$, $G_{max}$, and $\Delta$SD
may vary depending on specific lag times, results from the sensitivity analysis are shown in
Fig. 9. In general, $\Delta$SD showed the highest positive correlations with tracers and were most
sensitive for lag time of 1d, 5d, and 15d (Pearson correlation coefficient: 0.77, 0.63, and 0.85,
respectively; $p < 0.05$). Furthermore, regarding global solar radiation and maximum air
temperature, $G_{max1d}$ and $T_{max3d}$ showed best agreements (Pearson correlation coefficient: -0.83
and -0.7, respectively; $p < 0.05$).
To explore possible relationships between stream hydrochemistry ($\delta^2$H and EC) and nivo-
meteorological controls, selected indicators (at their most significant temporal scale) $T_{max3d}$,
$G_{max1d}$ and $\Delta$SD$_{15d}$ are shown in Fig.10 and 11. Those indicators represented the main drivers
of EC and $\delta^2$H variability within the Sulden and Trafoi catchment.
First, we observed that with increasing maximum air temperature $T_{max3d}$, EC concentration
clearly decreased, strongly influenced by the dilution effect of meltwater. For example, an
increase of $T_{max3d}$ by 5°C (from 0° to 5°C ) led to a decrease in EC in the Sulden and Trafoi
River by about 15 – 154 µS cm$^{-1}$ while a change from 10° to 15°C resulted in a drop of EC of
about 22 – 225 µS cm$^{-1}$ (Fig. 10a and b). Therefore, it can be noticed that the decrease in EC
was highest with relatively high $T_{max3d}$. Interestingly, the dilution seemed to depend also on
the sampling location along the stream and type of stream, as revealed by S6 (highest changes
in EC) and ST2 (lowest changes in EC) locations in the Sulden sub-catchment.
Secondly, we analysed the relationship of EC concentration and global solar radiation. As
shown in Fig. 10c to Fig. 10f, increasing maximum global solar radiation during the sampling





day $G_{max1d}$ (from 1400 to 1600 W m$^{-2}$) in the Sulden and Trafoi River led to strongly
decreased EC concentrations by about 94 – 382 µS cm$^{-1}$. In agreement with $T_{max3d}$, the highest
dilution effect was observed at S6. An isotopic depletion in δ$^2$H of 2.9‰ was calculated for
the Sulden River, while it notably was 7.1‰ for the Trafoi River.
Finally, we could explain the dilution effect also by the negative changes of snow depth ΔSD,
which represented the most sensitive variable to the temporal length (1d, 5d, and 15d)
compared to the other variables (Fig. 9). Using the example of ΔSD$_{15d}$ (measured at the
sampling day and 15 days prior to the sampling day), EC concentrations in both sub-
catchments resulted in less than 158 and 180 µS cm$^{-1}$ when losses of snow depths were about
50 to 70 cm (Trafoi and S1 – S4 streams, respectively). Smaller losses from 10 to 20 cm were
accompanied by still relatively high EC values of 256 and 301 µS cm$^{-1}$ (Trafoi and S1 – S4
streams, respectively) but led to a drop in EC concentrations by about 35 to 42 µS cm$^{-1}$ in
both sub-catchments. Therefore, the decrease in EC was highest with relatively high ΔSD$_{15d}$.

With respect to δ$^2$H, the dilution effect was associated with the typical isotopic depletion of
stream water, confirming the stream water dilution due to snowmelt input. On the one hand,
changes in snow depth from 60 to 50 cm of snow depth resulted in a depletion of 2.36 ‰ to
2.79 ‰ and 2.24 to 2.59 ‰ in δ$^2$H at Trafoi and Sulden (S1, S2, S5) streams, respectively. On
the other hand, changes of snow depth of less than 20 cm led only to smaller isotopic
depletion of 1.05 to 1.19 ‰ for the Trafoi and Sulden River. Not surprisingly, the clear linear
relationship between ΔSD and tracers held only for losses in snow depth. In contrast, positive
changes in ΔSD led to remarkably higher variability in EC and δ$^2$H in the river network.
**4    Discussion**
**4.1    Comparison of meteoric water lines**
The geographic origin of water vapour can generally be inferred by comparing the LMWL to
the GMWL (Craig 1961). Study results showed that precipitation was mainly formed by water
vapour originated from the Atlantic Ocean, which was in general agreement with the findings
of other studies. The LMWL of the Sulden catchment was very similar to the one from a
station at 2731 m a.s.l. in the Vermigliana Valley (δ$^2$H (‰)=8 δ$^{18}$O + 7.8) (Chiogna et al.,
2014) and a station at 2300 m a.s.l. in the Noce Bianco catchment (δ$^2$H (‰)=7.5 δ$^{18}$O + 7.9;





R² = 0.97, n=40) (Carturan et al., 2016), located south between the Ortles-Cevedale and
Adamello–Presanella group. However, it was slightly different in terms of d-excess when
considering the LMWL of Matsch/Mazia Valley (d-excess: 10.3, Penna et al., 2014) and
Northern Italy (d-excess: 9.4, Longinelli and Selmo, 2003). Moreover, it clearly differed from
the Mediterranean Meteoric Water Line (MMWL: $\delta^2H$ (‰) = 8 $\delta^{18}O$ + 22; Gat and Carmi,
1970). These observations may confirm the presence of different precipitation patterns and
microclimates at the regional scale (Brugnara et al., 2012).

## 4.2 Geological controls and hydrological connectivity

Geochemical dynamics were driven by a pronounced release of heavy metals (such as Al, V,
Cr, Ni, Zn, Cd, Pb) shown for the entire catchment and, in contrast, by a specific release of As
and Sr in the upper and lower Sulden sub-catchment (Fig. 3). Yet, as the explained variance
was only at about 53 %, further controls may be present. In this context, PC3 explained 11.8
% of additional variance and may represent surface vs. subsurface flows or residence time
within the soil.
With respect to the first observation, several sources of heavy metals can be addressed: on the
one hand, these elements may be released by rock weathering on freshly-exposed mineral
surfaces and sulphide oxidation, typically produced in metamorphic environments (Nordstrom
et al., 2011). Proglacial stream hydrochemistry may also strongly depend on the seasonal
evolution of the subglacial drainage system that contribute to specific element releases
(Brown and Fuge, 1998). In this context, rock glacier thawing may play an important role for
the release of Ni (Thies et al., 2007; Mair et al., 2011; Krainer et al., 2015) and Al and Mn
(Thies et al., 2013). However, high Ni concentrations were not observed in this study.
Moreover, high heavy metal concentrations were measured during the melting period in mid-
summer, which would generally be too early to derive from permafrost thawing (Williams et
al., 2006; Krainer et al., 2015). Also bedrock weathering as major origin probably needs to be
excluded because low concentrations occurred in winter when the hydrological connectivity at
higher elevations was still present (inferred from running stream water at the most upstream
locations).
On the other hand, it is therefore more likely that heavy metals derive from meltwater itself
due to the spatial and temporal dynamics observed. This would suggest that the element
release is strongly coupled with melting and infiltration processes, when hydrological



connectivity within the catchment is expected to be highest. To support this explanation,
supplementary element analysis of selected snowmelt (n = 2) and glacier melt (n = 2) samples
of this study were conducted. Although these samples did not contain high concentrations of
Cd, Ni, and Pb, for example, snowmelt in contact with the soil surface was more enriched in
such elements than dripping snowmelt. Moreover, snowmelt and ice melt samples from the
neighbouring Matsch/Mazia Valley in 2015 were strongly controlled by high Al, Co, Cd, Ni,
Pb and Zn concentrations (Engel et al., 2017). As shown for 21 sites in the Eastern Italian
Alps (Veneto and Trentino-South Tyrol region), hydrochemistry of the snowpack can largely
be affected by heavy metals originating from atmospheric deposition from traffic and industry
(such as V, Sb, Zn, Cd, Mo, and Pb) (Gabrielli et al., 2006). Likely, orographically induced
winds and turbulences arising in the Alpine valleys may often lead to transport and mixing of
trace elements during winter. Studies from other regions, such as Western Siberia Lowland
and the Tibetan Plateau, agree on the anthropogenic origin (Shevchenko et al., 2016 and Guo
et al., 2017, respectively).
In contrast, with respect to the origin of As and Sr, a clear geological source can be attributed,
supporting the first hypothesis on bedrock-specific geochemical signatures. In the lower
Sulden catchment (i.e. S1, S2, and T1), As could mainly originate from As-containing
bedrocks. As rich lenses are present in the cataclastic carbonatic rocks (realgar bearing) and in
the mineralized, arsenopyrite bearing bands of quartzphyllites, micaschists and paragneisses
of the crystalline basement. Different outcrops and several historical mining sites are known
and described in the literature (Mair, 1996, Mair et al., 2002, 2009; Stingl and Mair, 2005). In
the upper Sulden catchment, the presence of As is supported by the hydrochemistry of rock
glacier outflows in the Zay sub-catchment (corresponding to the drainage area of ST2; Engel
et al., 2018) but was not reported in other studies (Thies et al., 2007; Mair et al., 2011;
Krainer et al., 2015; Thies et al., 2013). Also high-elevation spring waters in the Matsch
Valley corroborated that As and Sr concentrations may originate from paragneisses and
micaschists (Engel et al., 2017). In this context, we suggest a controlling mechanism as
follows: the gradual decrease in As and Sr concentrations from rock glacier springs clearly
disagrees with the observations from other studies that rock glacier thawing in late summer
leads to increasing element releases (Williams et al., 2006; Thies et al., 2007; Krainer et al.,
2015; Nickus et al., 2015). Therefore, it is more likely that As and Sr originate from the
Quarzphyllite rocks, that form the bedrock of the rock glaciers (see Andreatta, 1952;



Montrasio et al., 2012). Weathering and former subglacial abrasion facilitate the release (Brown, 2002). As- and Sr-rich waters may form during winter when few quantities of water percolate in bedrock faults and then are released due to meltwater infiltration during summer (V. Mair, personal communication, 2018). As a clear delayed response of heavy metal concentrations in rock glacier outflow was revealed, the infiltration and outflow processes along flow paths in the bedrock near the rock glaciers may take up to two months to hydrochemically respond to snowmelt contamination.

As a consequence, a clear hydrochemical signature of permafrost thawing is difficult to find and results may lack the transferability to other catchments as not all rock glaciers contain specific elements to trace (Colombo et al., 2017). In this context, as precipitation and snowmelt affect the water budget of rock glaciers (Krainer and Mostler, 2002; Krainer et al., 2007), potential impacts of atmospheric inputs on rock glacier hydrochemistry could be assumed and would deserve more attention in future (Colombo et al., 2017).

Furthermore, export of elements in fluvial systems is complex and may strongly be affected by the pH (Nickus et al., 2015) or interaction with solids in suspension (Brown et al., 1996), which could not be addressed in this study. Further insights on catchment processes might be gained considering also element analysis of the solid fraction, to investigate whether water and suspended sediment share the same provenance.

## 4.3 The role of nivo-meteorological conditions and topography

Superimposing the impact of the geological origin, melting processes were controlled by meteorological conditions and topography, affecting stream hydrochemistry during summer, as shown by isotope dynamics (Fig. 5 and 7) and hydrochemical relationships (Fig. 8). It is well known that high correlations between snow or glacier melt and maximum air temperature exist (U.S. Army Corps of Engineers 1956; Braithwaite 1981), thus controlling daily meltwater contributions to streamflow (Mutzner et al., 2015; Engel et al., 2016). While ΔSD was used in this study, also snow depth and the extent of snow cover are suggested as effective indicators, exhibiting a strong control on runoff dynamics and thus melting processes (Singh et al., 2005). Likely, more specific explanatory variables such as vapour pressure, net radiation, and wind (Zuzel and Cox, 1975) or turbulent heat fluxes and long-wave radiation (Sicart et al., 2006) may exist but were not included in the present study due to the lack of observations.

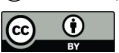



As shown in this study, dilutions effects and isotopic depletion could rather be explained by
maximum values $T_{max3d}$ and $G_{max1d}$ than averages of nivo-meteorological indicators or
precipitation-related indicators. This result confirms the second hypothesis on the importance
of nivo-meteorological indicators controlling melt processes by radiation and air temperature.
Such observation may imply the importance of threshold-like controls at the daily and short-
term scale, leading to tipping points along the cascade from atmospheric circulation and local
climate to hydrology to physico-chemical habitat (Milner et al., 2009). In this regard, the
(cumulated) daily maximum positive air temperature was used to characterize the decay of
simulated snow albedo related to snow metamorphism (Ragettli and Pellicciotti, 2012). The
authors also defined a threshold temperature for melt onset of 5°C, being in agreement with
our findings (shown in Fig. 7, Fig. 10a, and Fig. 10b). Moreover, relatively small changes and
low indicator values led to hydrochemical changes in stream water composition. This could
be justified by the fact that nivo-meteorological indicators were derived from 2825 m a.s.l.,
meaning that only about 30 % of the catchment area (assuming elevation bands of 50 m) were
above this location. Therefore, meteorological conditions and related nivo-meteorological
indicator may be more sensitive when compared to hydrochemical responses of the entire
catchment. While favourable melting conditions are certainly delayed at higher elevations,
stream water composition detected along the Sulden and Trafoi River (except S6 being closest
to the weather station) would mainly reflect melting processes originating from the lower
reaches within the catchment.
In this study, the most pronounced dilution effect and isotopic depletion (regarding monthly
data) could be attributed to $G_{max1d}$, which thus may be considered as the most relevant nivo-
meteorological indicator. This observation could be supported by Vincent and Six (2013),
who found that spatial variations of ice ablation were mainly driven by potential solar
radiation. It is further considered to be the main energy source driving melt processes in
glacierized catchments of different climates (Sicart et al., 2008) and may integrate the effect
of cloud coverage (Anslow et al., 2008). In contrast, lower radiation inputs and subzero air
temperatures occurred during snowfall events (indicated by positive ΔSD) and likely
interrupted melt processes, leading to higher variability of hydrochemical stream water
composition (Hannah et al., 1999; Sicart et al., 2006; DeBeer and Pomeroy, 2010).
Results from the temporal sensitivity analysis are generally difficult to compare due to the
lack of suitable studies and thus provide a novel data set for glacierized catchments. The



sensitivity of ΔSD to different temporal length (3d, 5d, 15d) may indicate potential meltwater
storage components and their effectiveness to route meltwater at different temporal scales.
First, the snowpack represents a short-term storage for meltwater ranging from few hours to
few days (Coléou and Lesaffre, 1998), due to different snowpack properties (i.e. irreducible
water saturation, layer thickness) (Colbeck 1972; Marsh and Pomeroy, 1996). Second, the
presence of slower and quicker flow paths within glacial till, talus, moraines, and shallow vs.
deeper groundwater compartments could justify the intermediate (5d) and longer (15d)
meltwater response (Brown et al., 2006; Roy and Hayashi, 2009; McClymont et al., 2010;
Fischer et al., 2015; Weiler et al., 2017).

### 623    4.4   Implications for streamflow and hydrochemistry dynamics

Tracer dynamics of EC and stable isotopes associated with monthly discharge variations
generally followed the conceptual model of the seasonal evolution of streamflow
contributions, as described for catchments with glacierized area of 17 % (Penna et al. 2017)
and 30 % (Schmieder et al. 2017). However, isotopic dynamics were generally less
pronounced compared to these studies, likely resulting from the impact of relative meltwater
contribution related to different catchment sizes and the proportion of glacierized area (Baraer
et al., 2015).
In addition, hydrometric and geochemical dynamics analysed in this study were controlled by
an interplay of meteorological conditions and the heterogeneity of geology. Such an interplay
is highlighted by EC dynamics (i.e. EC variability derived from VC), to be further controlled
by the contributing catchment area (i.e. EC gradients along the Sulden and Trafoi River). As
EC was highly correlated to Ca concentration (Spearman rank correlation: 0.6, $p < 0.05$; see
Fig. 4), EC dynamics were determined by the spatial distribution of different geology. For
example, as dolomitic rocks are present almost within the entire Trafoi sub-catchment,
meltwater following the hydraulic gradient can likely become more enriched in solutes with
longer flow pathways and increasing storage capability related to the catchment size (Fig. 6).
As consequence, the 'EC enrichment gradient' could persist during both the melting period
and baseflow conditions in the presence of homogenous geology. Therefore, topography as
control may become more important than the geological setting, to control spatial stream
water variability. In the Sulden sub-catchment, however, dolomitic rocks are only present in
the upper part of the catchment while metamorphic rocks mostly prevail. This leads to a



pronounced dilution of Ca-rich waters with increasing catchment area or in other words,
increasing distance from the source area (Fig. 6) during baseflow conditions. This implies that
meltwater contributions to the stream homogenize the effect of geographic origin on different
water sources, having the highest impact in vicinity to the meltwater source (see Table 6).
The additional effect of topographical characteristics is underlined by the findings that the
Sulden River hydrochemistry at S2 was significantly more depleted in $\delta^2$H and $\delta^{18}$O than T1
hydrochemistry. Compared with the Sulden sub-catchment, the Trafoi sub-catchment has a
slightly lower proportion of glacier extent but, more importantly, has a clearly smaller
catchment area within the elevation bands of 1800 to 3200 m a.s.l. (i.e. 40.2 km² for the
Trafoi and 66.5 km² for the Sulden sub-catchment). In this elevation range, the sub-
catchments of major tributaries ST1, ST2, and ST3 are situated, which deliver large snowmelt
contributions to the Sulden River (Fig. 6).
In consequence, resulting from the impact of these different controls, specific hydrometric
and hydrochemical relationships derive. For example, the hysteretic relationship between
discharge and EC (Fig. 8b) helps to identify the conditions with maximum discharge and EC:
during baseflow conditions, the Sulden River showed highest EC of about 350 µS cm$^{-1}$
seemingly to be bound to only about 3 m³ s$^{-1}$ whereas the maximum dilution effect occurred
during a storm on 29 June 2014 (55 mm of precipitation at AWS Madritsch) with 29.3 m³ s$^{-1}$
of discharge resulting in only 209 µScm$^{-1}$. However, these observations based on daily data
sampled at 23:00, likely not capturing the entire hydrochemical variability inherent of the
Sulden catchment. As shown in Fig. 5 and Fig. 7, much higher discharges and thus even lower
EC could be reached along the Sulden River and inversely, which was potentially limited by
the specific geological setting of the study area.
As more extreme weather conditions (such as heat waves, less solid winter precipitation) are
expected in future (Beniston, 2003; Viviroli et al., 2011; Beniston and Stoffel 2014),
glacierized catchments may exhibit more pronounced hydrochemical responses such as
shifted or broader ranges of hydrochemical relationships and increased heavy metal
concentrations both during melting periods and baseflow conditions. However, identifying
these relationships with changing meteorological conditions would deserve more attention
and is strongly limited by our current understanding of underlying hydrological processes
(Schaefli et al., 2007). In a changing cryosphere, more complex processes such as non-
stationarity processes may emerge under changing climate, which itself was found to be a



major cause of non-stationarity (Milly et al., 2008). In this context, explaining the hydrochemical dynamics ambiguity observed during the baseflow period in November 2015 (Fig. 7) will deserve further attention.

Finally, our results can partly confirm the third hypothesis following Heidbüchel et al. (2013). Long-term controls such as geology and topography govern hydrochemical responses at the spatial scale (such as bedrock-specific geochemical signatures, EC gradients, and relative snowmelt contribution). In contrast, short-term controls such as maximum daily solar radiation, air temperature, and snow depth differences drive short-term responses (such as discharge variability and EC dilution). However, as the catchment response strongly depended on the melting period vs. baseflow conditions, controls at longer temporal scales interact as well. Thus, our findings suggest that glacierized catchments react in a much more complex way and that catchment responses cannot be attributed to one specific scale, justified by either short-term or long-term controls alone.

In this context, the present study provides novel insights into geological, meteorological, and topographic controls of stream water hydrochemistry rarely addressed for glacierized catchments so far. Moreover, this study strongly capitalizes on an important dataset that combines nivo-meteorological indicators and different tracers (stable isotopes of water, EC, major, minor and trace elements), underlining the need for conducting multi-tracer studies in complex glacierized catchments.

### 4.5   Methodological limitation

The sampling approach combined a monthly spatial sampling with daily sampling at the outlet, which methodologically is in good agreement with other sampling approaches, accounting for increasing distance of sampling points to the glacier (Zhou et al., 2014; Baraer et al., 2015), intense spatial and temporal sampling (Penna et al., 2014; Fischer et al., 2015), synoptic sampling (Carey et al., 2013; Gordon et al., 2015), and different catchment structures such as nested catchments (Soulsby et al., 2006b). Sampling covered a variety of days with typical snowmelt, glacier melt and baseflow conditions during 2014 and 2015, confirming the representativeness of tracer dynamics within two years contrasting in their meteorological characteristics (Table 1). However, short-term catchment responses (such as storm-induced peak flows and related changes in hydrochemistry) were difficult to be captured by this sampling approach. Furthermore, two years of field data are probably not sufficient to capture




all hydrological conditions and catchment responses to specific meteorological conditions. In this regards, long-term studies may have better chances in capturing the temporal variability of hydrochemical responses (Thies et al., 2007). In this context, sampling approaches might need to become more complex in future to unravel further process understanding of glacierized catchments.

## 5 Conclusions

Our results highlight the complex hydrochemical responses of mountain glacierized catchments at different temporal and spatial scales. To our knowledge, only few studies investigated the impact of controlling factors on stream water hydrochemistry by using nivo-meteorological indicators and multi-tracer data, which we recommend to establish as prerequisite for studies in other glacierized catchments.

The main results of this study can be summarized as follows:

- Hydrometric and geochemical dynamics were controlled by an interplay of meteorological conditions and the geological heterogeneity. The majority of the variance (PC1: 36.3 %) was explained by heavy metal concentrations (such as Al, V, Cr, Ni, Zn, Cd, Pb), associated with atmospheric deposition on the snowpack and release through snowmelt. Remaining variance (PC2: 16.3 %) resulted both from the presence of a bedrock-specific geochemical signature (As and Sr concentrations) and the role of snowmelt contribution.

- The isotopic composition of rock glacier outflow was relatively similar to the composition of glacier melt whereas high concentrations of As and Sr may more likely result from bedrock weathering.

- At the monthly scale for different sub-catchments (spatial scale: 0.05 – 130 km²), both $\delta^{18}$O and EC revealed complex spatial and temporal dynamics such as contrasting EC gradients during baseflow conditions and melting periods.

- At the daily scale for the entire study area (spatial scale: 130 km²), we observed strong relationships of hydrochemical variables, with mainly discharge and EC exhibiting a strong monthly relationship. This was characterised by a hysteretic-like pattern, determined by highest EC and lowest discharge during baseflow conditions on the one hand and maximum EC dilution due to highest discharge during a summer storm.





• Main drivers of EC and $\delta^2$H variability were the nivo-meteorological indicators $T_{max3d}$,

$G_{max1d}$ and $\Delta SD_{15d}$. $\Delta SD$ was found to be the most sensitive variable to different

temporal lengths (3d, 5d, and 15d) and $G_{max1d}$ resulted in the most pronounced EC

dilution and isotopic depletion.

Finally, this study may support future classifications of glacierized catchments according to
their hydrochemical response under different catchment conditions or the prediction of
appropriate end-member signatures for hydrograph separation being valid at longer time
scales.
**6    Data availability**
Hydrometeorological data are available upon request at the Hydrographic Office of the
Autonomous Province of Bozen-Bolzano. Tracer data used in this study are freely available
by contacting the authors.

**7    Acknowledgements**
This research is part of the GLACIALRUN project and funded by the foundation of the Free
University of Bozen-Bolzano and supported by the project "Parco Tecnologico - Tecnologie
ambientali".
The authors thank Andrea Ruecker, Alex Boninsegna, Raffael Foffa, and Michiel Blok for
their field assistance. Giulia Zuecco and Luisa Pianezzola are thanked for the isotopic analysis
at TESAF, University of Padova and Christian Ceccon for the isotopic analysis in the
laboratory of the Free University of Bozen-Bolzano. We also thank Giulio Voto at
EcoResearch s.r.l. (Bozen/Bolzano) for the element analysis. We appreciate the helpful
support for the geological interpretation by Volkmar Mair. We acknowledged the project
AQUASED, whose instrumentation infrastructure we could use. Furthermore, we thank the
Hydrographic Office and the Department of Hydraulic Engineering of the Autonomous
Province of Bozen-Bolzano for providing meteorological and hydrometric data. We
acknowledge the Forestry Commission Office Prat, the National Park Stilfser Joch / Passo
Stelvio, and the Cable car Sulden GmbH for their logistical support and helpful advices.



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



Table 1. Meteorological characteristics of the weather station Madritsch/Madriccio 2.825 m
a.s.l. in 2014 and 2015.

| Date | 2014 | 2015 |
| --- | --- | --- |
| Precipitation (total / rain / snow) (mm y$^{-1}$)* | 1284/704/579 | 961/637/323 |
| Mean annual air temperature (°C) | -1.4 | -0.8 |
| Days with snow cover > 10cm | 270 | 222 |
| Maximum snow depth (date) | 02/03/2014 | 27/03/2015 |
| Maximum snow depth (cm) | 253 | 118 |
| Date of snow cover disappearance | 12/07/2014 | 13/06/2015 |
| Average discharge (median) (m³ s$^{-1}$) | 9.5 | 5.2 |

* Precipitation data are not wind-corrected. Rain vs. snow separation was performed
following Auer (1974)




Table 2. Topographical characteristics of sub-catchments defined by sampling points.

| Sampling point | Description | Catchment area (km²) | Glacier cover (%) | Elevation range |
|---|---|---|---|---|
| T1 | Trafoi River | 12.18 | 17 | 1197 - 3889 |
| T2 | Trafoi River | 46.72 | 18.6 | 1404 - 3889 |
| T3 | Trafoi River | 51.28 | 35 | 1587 - 3469 |
| TT1 | Tributary draining Trafoi glacier | 4.32 | 27.1 | 1587 - 3430 |
| TT2 | Small creek | 0.05 | 0 | 1607 - 2082 |
| TT3 | Tributary draining Zirkus/ Circo glacier | 6.46 | 44 | 1605 - 3888 |
| TSPR1 | Spring at the foot of a slope | - | 0 | 1602* |
| TSPR2 | Spring at the foot of a slope | - | 0 | 1601* |
| S1 | Sulden River | 130.14 | 13.6 | 1109 - 3896 |
| S2 | Sulden River | 74.61 | 12.1 | 1296 - 3896 |
| S3 | Sulden River | 57.01 | 15.8 | 1707 - 3896 |
| S4 | Sulden River | 45.06 | 18.6 | 1838 - 3896 |
| S5 | Sulden River | 18.91 | 29.7 | 1904 - 3896 |
| S6 | Sulden River | 14.27 | 38.5 | 2225 - 3896 |
| ST1 | Razoi tributary | 6.46 | 0.6 | 1619 - 3368 |
| ST2 | Zay tributary | 11.1 | 12.8 | 1866 - 3543 |
| ST3 | Rosim tributary | 7.3 | 9.7 | 1900 - 3542 |



| SSPR1 | Spring in the valley bottom near Sulden town | - | 0 | 1841* |
| SSPR2 - 4 | At the base of the rock glacier front | - | 0.12** | 2614, 2594, 2600* |

* for spring locations, the elevation of the sampling point is given.
** for rock glacier spring locations, the glacier cover refers to the extent of both rock glaciers.

Table 3. Environmental variables derived from the weather station Madritsch/Madriccio at
2825 m a.s.l..

| Variable | Unit | Description |
|---|---|---|
| $P_{1d}$ | mm | Cumulated precipitation of the sampling day |
| $P_{nd}$ | | Cumulated precipitation n days prior to sampling day |
| $T_{max1d}$ | °C | Maximum air temperature during the sampling day |
| $T_{maxnd}$ | | Maximum air temperature within n days prior to sampling day |
| $G_{max1d}$ | W/m² | Maximum global solar radiation during sampling day |
| $G_{maxnd}$ | | Maximum global solar radiation within n days prior to sampling day |
| $\Delta SD_{1d}$ | cm | Difference of snow depth measured at the sampling day at 12:00 and the previous day at 12:00, based on 6h averaged snow depth records. |
| $\Delta SD_{nd}$ | | Difference of snow depth measured at the sampling day at 12:00 and n days prior the sampling day at 12:00, based on 6h averaged snow depth records. |
| $D_{Prec1}$ | days | Days since last daily cumulated precipitation of > 1mm was measured. |
| $D_{Prec10}$ | | Days since last daily cumulated precipitation of > 10mm was measured. |
| $D_{Prec20}$ | | Days since last daily cumulated precipitation of > 20mm was measured. |





Table 4. Statistics of element concentration (in µg l⁻¹) from selected stream, tributary and active rock glacier springs in the Sulden catchment sampled from March to October 2015. CV: coefficient of variation. VC: variability coefficient (see Eq. 1) with $SD_{baseflow}$ (based on samples from March, April, and October 2015) and $SD_{melting}$ (based on samples from May to September 2015). Note that CV was not calculated for SSPR2 – 4 as water samples were available only during summer.

| Location | Statistic | Na | Mg | Al | K | Ca | V | Cr | Mn | Fe | Ni | Cu |
|---|---|---|---|---|---|---|---|---|---|---|---|---|
| S1 | min | 1881.3 | 12169.1 | 6.9 | 1051.2 | 41497.2 | 0.2 | 0.2 | 1.1 | 21.1 | 0.5 | 1.5 |
| | max | 7246.9 | 19547.1 | 541.4 | 2456.0 | 56508.3 | 1.8 | 1.4 | 62.4 | 1038.9 | 3.8 | 9.1 |
| | mean | 3253.5 | 14625.4 | 148.7 | 1657.3 | 48423.7 | 0.6 | 0.6 | 15.0 | 292.5 | 1.3 | 4.9 |
| | SD | 1782.0 | 2265.3 | 157.3 | 487.1 | 4538.1 | 0.5 | 0.3 | 18.7 | 300.2 | 1.0 | 3.0 |
| | CV | 0.5 | 0.2 | 1.1 | 0.3 | 0.1 | 0.9 | 0.5 | 1.2 | 1.0 | 0.8 | 0.6 |
| | VC | 0.6 | 0.3 | 0.3 | 1.6 | 0.5 | 0.2 | 0.2 | 0.1 | 0.3 | 0.2 | 0.8 |
| S2 | min | 1968.4 | 9793.3 | 6.1 | 1546.3 | 43167.9 | 0.1 | 0.2 | 1.1 | 12.0 | 0.3 | 1.3 |
| | max | 3334.6 | 16453.8 | 743.1 | 2476.3 | 73177.3 | 1.9 | 1.7 | 71.0 | 1513.5 | 3.8 | 9.1 |
| | mean | 2431.6 | 12437.2 | 211.2 | 1900.9 | 52361.7 | 0.6 | 0.6 | 18.5 | 410.7 | 1.2 | 3.3 |
| | SD | 409.4 | 2292.5 | 236.4 | 299.3 | 8738.1 | 0.6 | 0.5 | 22.4 | 467.9 | 1.1 | 2.4 |





| Group | Stat | | | | | | | | | | | |
|---|---|---|---|---|---|---|---|---|---|---|---|---|
|  | CV | 0.2 | 0.2 | 1.1 | 0.2 | 0.2 | 1.0 | 0.8 | 1.2 | 1.1 | 0.9 | 0.7 |
|  | VC | 2.0 | 0.2 | 0.2 | 0.7 | 0.2 | 0.1 | 0.2 | 0.1 | 0.2 | 0.2 | 0.2 |
| S6 | min | 1262.6 | 17458.6 | 9.0 | 1042.6 | 67588.1 | 0.1 | 0.1 | 1.5 | 21.6 | 0.5 | 1.5 |
|  | max | 2277.0 | 34928.5 | 799.4 | 1748.4 | 166731.5 | 3.4 | 1.9 | 104.6 | 1587.1 | 6.2 | 17.0 |
|  | mean | 1805.6 | 22862.4 | 278.4 | 1362.7 | 129896.0 | 1.1 | 0.8 | 43.1 | 596.1 | 2.1 | 6.5 |
|  | SD | 339.4 | 5512.9 | 321.0 | 259.4 | 28165.0 | 1.2 | 0.7 | 47.4 | 670.0 | 1.9 | 4.9 |
|  | CV | 0.2 | 0.2 | 1.2 | 0.2 | 0.2 | 1.2 | 0.8 | 1.1 | 1.1 | 0.9 | 0.8 |
|  | VC | 0.6 | 0.2 | 0.0 | 1.4 | 0.5 | 0.0 | 0.1 | 0.0 | 0.1 | 0.1 | 0.2 |
| SSPR2-4 | min | 1768.3 | 10051.4 | 9.0 | 1236.1 | 76848.5 | 0.0 | 0.1 | 1.5 | 16.7 | 0.2 | 0.5 |
|  | max | 2818.6 | 29509.5 | 321.2 | 2402.5 | 131149.7 | 2.5 | 0.6 | 71.7 | 492.2 | 1.5 | 38.3 |
|  | mean | 2199.9 | 17254.4 | 68.9 | 2009.0 | 94611.4 | 0.4 | 0.3 | 13.1 | 127.5 | 0.7 | 8.2 |
|  | SD | 343.3 | 6935.8 | 97.8 | 294.4 | 21508.4 | 0.8 | 0.2 | 22.5 | 148.5 | 0.5 | 11.7 |
|  | CV | 0.2 | 0.4 | 1.4 | 0.1 | 0.2 | 2.2 | 0.5 | 1.7 | 1.2 | 0.7 | 1.4 |
| T1 | min | 1125.7 | 13481.8 | 6.3 | 536.9 | 33044.0 | 0.2 | 0.1 | 0.9 | 13.3 | 0.3 | 0.4 |





|  |  | | | | | | | | | | |
|---|---|---|---|---|---|---|---|---|---|---|---|
|  | max | 3312.9 | 42197.2 | 914.7 | 1470.6 | 88033.8 | 4.5 | 1.8 | 121.8 | 1178.5 | 3.5 | 22.0 |
|  | mean | 2078.3 | 19230.5 | 139.8 | 985.9 | 48369.3 | 0.8 | 0.5 | 19.1 | 190.2 | 1.1 | 5.1 |
|  | SD | 600.5 | 8846.6 | 293.5 | 302.7 | 16108.6 | 1.4 | 0.5 | 38.9 | 374.8 | 1.0 | 6.6 |
|  | CV | 0.3 | 0.5 | 2.1 | 0.3 | 0.3 | 1.8 | 1.0 | 2.0 | 2.0 | 0.9 | 1.3 |
|  | VC | 1.3 | 0.1 | 0.0 | 0.8 | 0.3 | 0.0 | 0.3 | 0.0 | 0.0 | 0.2 | 0.2 |
| TT2 | min | 321.0 | 12048.8 | 4.7 | 272.8 | 23873.4 | 0.1 | 0.2 | 0.8 | 10.4 | 0.3 | 0.7 |
|  | max | 2524.5 | 20756.5 | 568.0 | 1017.1 | 39335.1 | 2.0 | 1.3 | 57.1 | 1116.2 | 2.7 | 22.2 |
|  | mean | 1148.1 | 16898.0 | 97.0 | 551.6 | 32228.7 | 0.4 | 0.4 | 10.2 | 173.2 | 0.9 | 8.0 |
|  | SD | 727.9 | 2945.5 | 179.7 | 244.1 | 4615.5 | 0.6 | 0.4 | 17.9 | 357.5 | 0.7 | 7.7 |
|  | CV | 0.6 | 0.2 | 1.9 | 0.4 | 0.1 | 1.5 | 0.9 | 1.8 | 2.1 | 0.8 | 1.0 |
|  | VC | 0.9 | 0.8 | 0.1 | 0.6 | 0.5 | 0.1 | 0.3 | 0.1 | 0.1 | 0.3 | 0.2 |





Table 5. Statistics of element concentration (in µg l$^{-1}$) from selected stream, tributary and active rock glacier springs in the Sulden catchment
sampled from March to October 2015. CV: coefficient of variation. VC: variability coefficient (see Eq. 1) with SD$_{baseflow}$ (based on samples
from March, April, and October 2015) and SD$_{melting}$ (based on samples from May to September 2015). Note that CV was not calculated for
SSPR2 – 4 as water samples were available only during summer.

| location | statistics | Zn | As | Se | Rb | Sr | Ag | Cd | Sb | Hg | Pb | U |
|---|---|---|---|---|---|---|---|---|---|---|---|---|
| S1 | min | 4.1 | 12.1 | 0.5 | 0.0 | 307.9 | 0.0 | 0.0 | 0.2 | 0.0 | 0.4 | 0.0 |
|  | max | 23.2 | 61.1 | 1.1 | 2.6 | 390.5 | 0.1 | 0.1 | 0.5 | 0.2 | 7.6 | 11.3 |
|  | mean | 9.7 | 27.0 | 0.8 | 1.1 | 349.8 | 0.0 | 0.1 | 0.3 | 0.1 | 2.1 | 5.1 |
|  | SD | 5.8 | 15.5 | 0.2 | 1.1 | 27.2 | 0.0 | 0.1 | 0.1 | 0.1 | 2.3 | 5.2 |
|  | CV | 0.6 | 0.6 | 0.2 | 1.0 | 0.1 | 2.6 | 1.0 | 0.4 | 1.1 | 1.1 | 1.0 |
|  | VC | 0.2 | 2.6 | 1.0 | 0.0 | 0.7 | - | 1.0 | 2.0 | 0.0 | 0.1 | 0.0 |
| S2 | min | 3.7 | 15.1 | 0.4 | 0.0 | 334.0 | 0.0 | 0.0 | 0.1 | 0.0 | 0.3 | 0.0 |
|  | max | 23.8 | 40.9 | 0.7 | 3.4 | 609.9 | 0.0 | 0.1 | 0.2 | 0.2 | 9.4 | 11.3 |
|  | mean | 8.5 | 23.3 | 0.5 | 1.6 | 410.7 | 0.0 | 0.0 | 0.2 | 0.1 | 2.7 | 4.9 |
|  | SD | 6.4 | 8.0 | 0.1 | 1.6 | 81.0 | 0.0 | 0.0 | 0.0 | 0.1 | 3.4 | 5.1 |



| | | | | | | | | | | | |
|---|---|---|---|---|---|---|---|---|---|---|---|
| CV | 0.7 | 0.3 | 0.2 | 1.0 | 0.2 | - | 1.3 | 0.3 | 1.1 | 1.3 | 1.0 |
| VC | 0.2 | 2.0 | 0.5 | 0.0 | 0.3 | - | 1.0 | 1.0 | 0.0 | 0.1 | 0.0 |
| S6 min | 5.6 | 6.3 | 0.5 | 0.0 | 524.0 | 0.0 | 0.0 | 0.3 | 0.0 | 0.4 | 0.0 |
| max | 40.9 | 17.0 | 1.2 | 1.9 | 2024.0 | 0.0 | 0.2 | 0.5 | 0.1 | 18.1 | 11.3 |
| mean | 19.1 | 10.1 | 0.9 | 0.7 | 1380.5 | 0.0 | 0.1 | 0.3 | 0.0 | 6.7 | 4.0 |
| SD | 12.9 | 4.0 | 0.2 | 0.8 | 463.1 | 0.0 | 0.1 | 0.1 | 0.0 | 7.3 | 4.9 |
| CV | 0.7 | 0.4 | 0.2 | 1.2 | 0.3 | - | 0.9 | 0.2 | 1.2 | 1.1 | 1.2 |
| VC | 0.2 | 0.1 | 0.5 | 0.0 | 0.5 | - | 0.5 | 2.2 | 0.0 | 0.0 | 0.0 |
| SSPR2-4 min | 1.5 | 6.3 | 0.4 | 0.0 | 341.2 | 0.0 | 0.0 | 0.1 | 0.0 | 0.2 | 0.0 |
| max | 49.4 | 38.0 | 0.6 | 2.7 | 1355.7 | 0.1 | 0.4 | 0.4 | 0.1 | 19.8 | 27.2 |
| mean | 10.7 | 31.1 | 0.5 | 0.9 | 770.9 | 0.0 | 0.1 | 0.2 | 0.0 | 3.1 | 6.9 |
| SD | 14.8 | 4.4 | 0.1 | 1.0 | 435.7 | 0.0 | 0.1 | 0.1 | 0.0 | 6.3 | 9.4 |
| CV | 1.4 | 0.1 | 0.2 | 1.1 | 0.6 | 2.6 | 1.4 | 0.6 | 1.3 | 2.0 | 1.4 |





| T1 | | | | | | | | | | | |
|---|---|---|---|---|---|---|---|---|---|---|---|
| min | 2.3 | 7.2 | 0.6 | 0.0 | 220.9 | 0.0 | 0.0 | 0.2 | 0.0 | 0.3 | 0.0 |
| max | 46.5 | 64.2 | 1.4 | 1.9 | 478.1 | 0.0 | 0.2 | 0.7 | 0.2 | 18.0 | 12.5 |
| mean | 10.9 | 24.5 | 1.1 | 0.7 | 340.1 | 0.0 | 0.1 | 0.4 | 0.1 | 2.9 | 5.6 |
| SD | 13.6 | 18.4 | 0.3 | 0.7 | 75.8 | 0.0 | 0.1 | 0.1 | 0.1 | 5.7 | 5.7 |
| CV | 1.2 | 0.8 | 0.2 | 1.1 | 0.2 | - | 1.4 | 0.4 | 1.1 | 2.0 | 1.0 |
| VC | 0.1 | 2.9 | 0.6 | 0.0 | 0.9 | - | 0.6 | 2.0 | 0.0 | 0.0 | 0.0 |
| TT2 | | | | | | | | | | | |
| min | 2.8 | 0.3 | 0.5 | 0.0 | 149.4 | 0.0 | 0.0 | 0.2 | 0.0 | 0.3 | 0.0 |
| max | 39.4 | 1.2 | 1.5 | 1.7 | 384.5 | 0.5 | 0.1 | 0.5 | 0.7 | 9.1 | 10.6 |
| mean | 9.9 | 0.7 | 1.0 | 0.4 | 247.5 | 0.1 | 0.0 | 0.3 | 0.1 | 1.8 | 4.8 |
| SD | 11.4 | 0.3 | 0.3 | 0.5 | 67.5 | 0.2 | 0.0 | 0.1 | 0.2 | 2.8 | 4.9 |
| CV | 1.2 | 0.4 | 0.3 | 1.5 | 0.3 | 2.6 | 1.3 | 0.4 | 1.8 | 1.5 | 1.0 |
| VC | 0.1 | 0.3 | 1.3 | 0.0 | 1.2 | 0.0 | 1.0 | - | 0.0 | 0.1 | 0.0 |




1 Table 6. Variability coefficient (VC) for selected locations along the Sulden and Trafoi River

2 in 2014 and 2015.

| Location | River section (in km) | VC |
|----------|-----------------------|------|
| T3 | 6.529 | 0.70 |
| T2 | 2.774 | 0.85 |
| T1 | 51 | 1.09 |
| S6 | 12.87 | 0.01 |
| S3 | 6.417 | 0.42 |
| S2 | 2.739 | 0.35 |
| S1 | 0 | 0.77 |



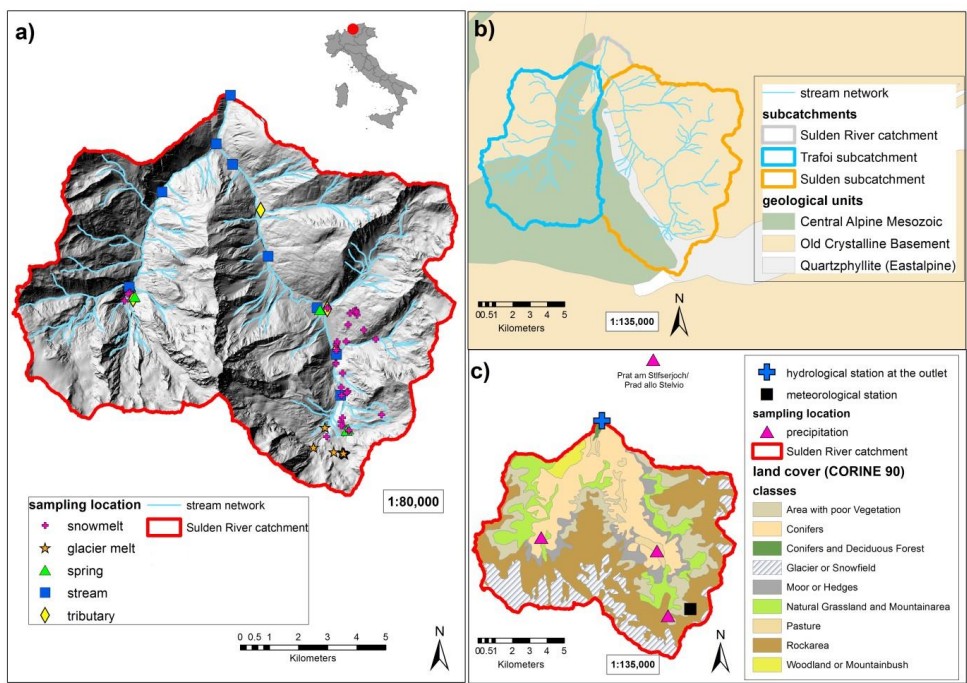

**Figure 1. Overview of the Sulden catchment with a) sampling point, b) geology, and c) land cover with**

**instrumentation. The meteorological station shown is the Madritsch/Madriccio AWS of the Hydrographic Office**

**(Autonomous Province of Bozen-Bolzano). The glacier extent refers to 2006 (Autonomous Province of Bozen-**

**Bolzano).**









**Figure 2. Meteoric water line of different water sources sampled in the Sulden catchment in 2014 and 2015. The inset**
**shows a zoom on rainfall, snow, snowmelt, glacier melt, and spring waters with the regression line of snowmelt**
**samples collected from spring to autumn.**

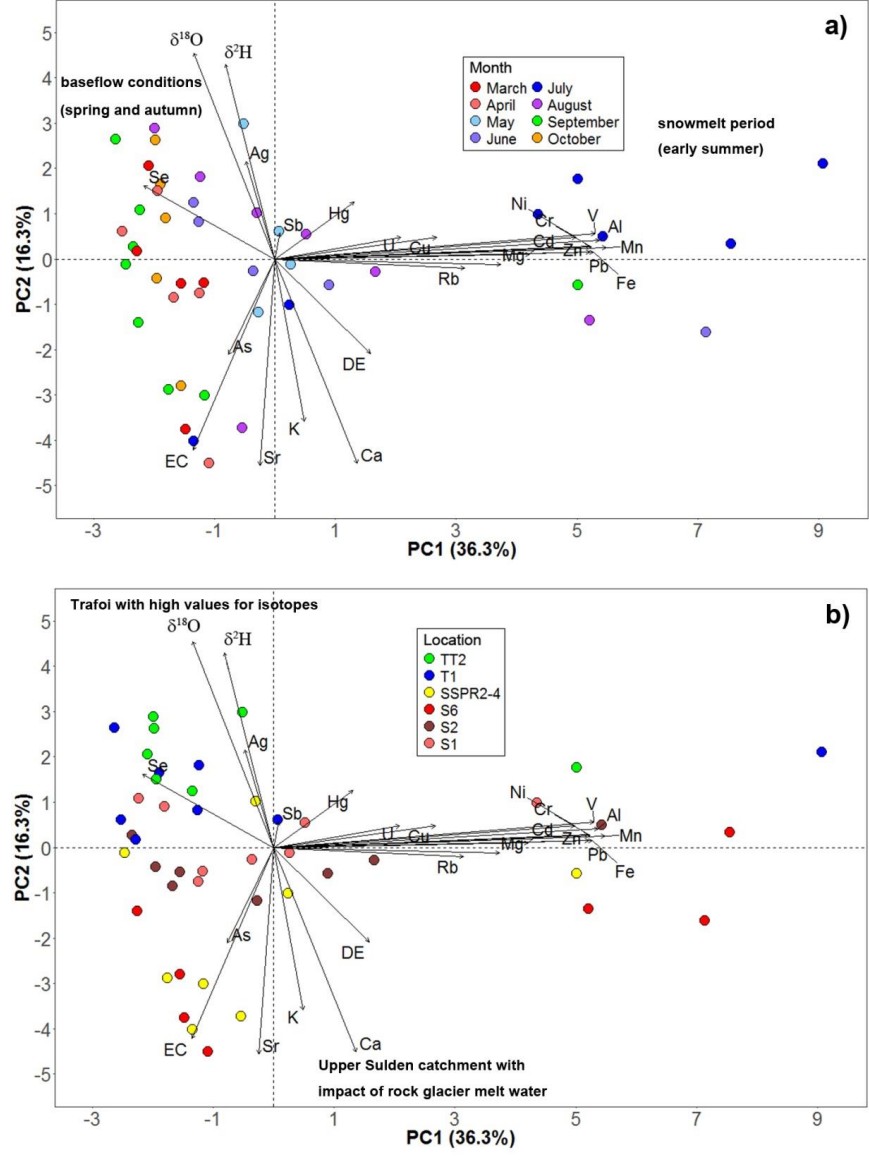

**Figure 3. Principle component analysis of element concentrations of stream water and springs draining a rock glacier**
**sampled in the Sulden and Trafoi sub-catchments from March to October 2015. Data based on n = 47 samples are**
**shown in groups according to a) the sampling locations and b) the sampling month.**





**Figure 4. Spearman rank correlation matrix of hydrochemical variables. Values are shown for a level of significance *p***

**< 0.05.**





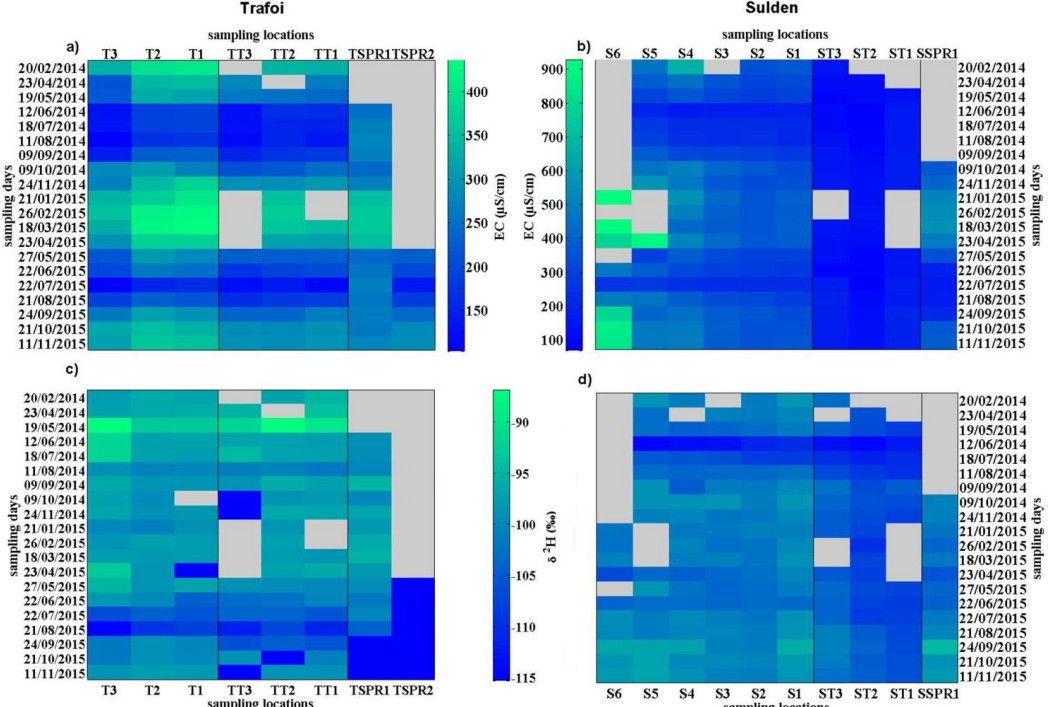

**Figure 5. Spatial and temporal variability of EC (µS cm⁻¹) and δ²H (‰) at different stream sections, tributaries and**
**springs within the Trafoi sub-catchment (subplot a and c) and the Sulden sub-catchment (subplot b and d) in 2014**
**and 2015. The heatmaps are grouped into locations at streams, tributaries, and springs. Grey areas refer to missing**
**sample values due to frozen or dried out streams/tributaries or because the sampling location was included later in the**
**sampling scheme.**



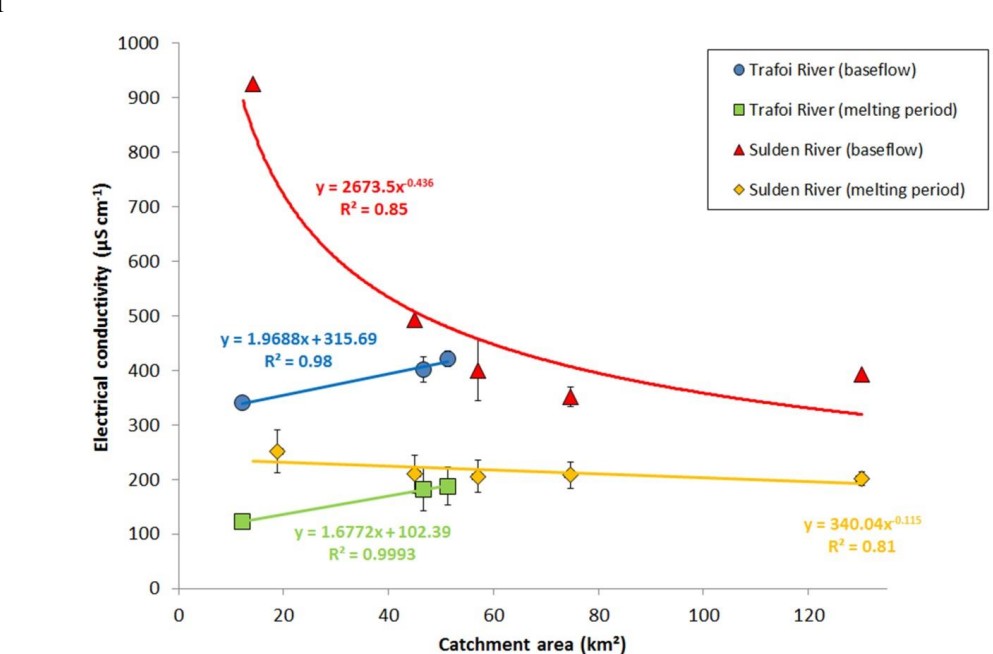

3  **Figure 6. Spatial variability of electrical conductivity along the Trafoi and Sulden River against catchment area.**

4  **Electrical conductivity is averaged for sampling days during baseflow conditions (21/01/2015, 26/02/2015, and**



**18/03/2015)     and     melt     period     (12/06/2014,     18/07/2014,     11/08/2014,     and     09/09/2014).**

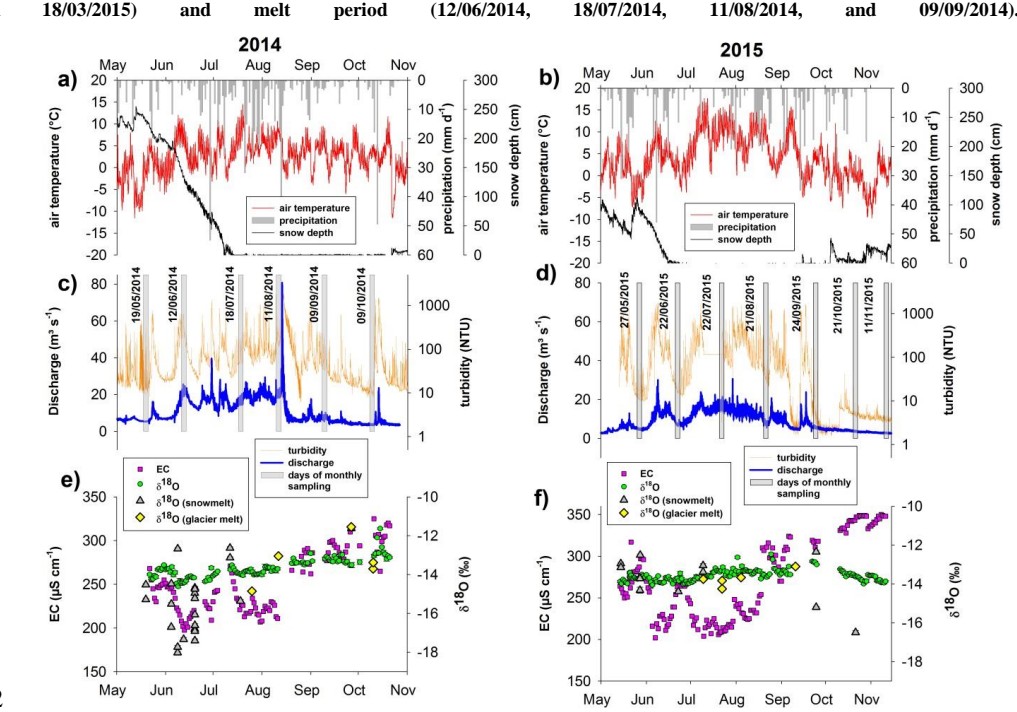

**Figure 7. Time series from 2014 and 2015 of a) and b) precipitation, hourly air temperature and snow depth at the**
**AWS Madritsch, c) and d) streamflow and turbidity, e) and f) electrical conductivity and δ¹⁸O of the stream at the**
**outlet station Ponte Stelvio and of snowmelt and glacier melt water. Grey shaded bars indicate the date of monthly**
**sampling carried out in the entire catchment.**



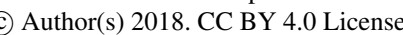


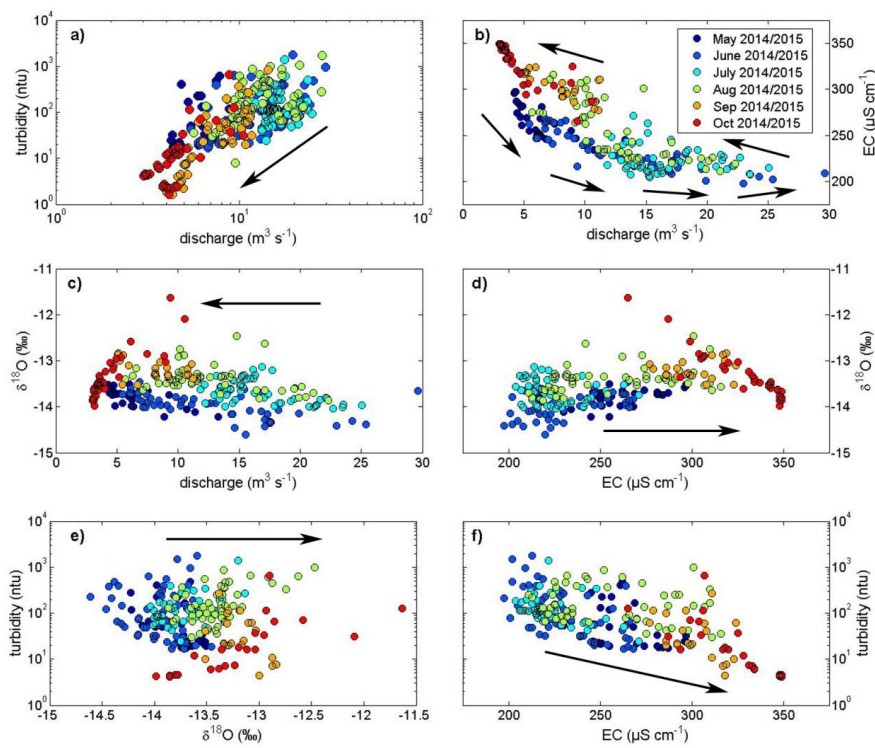

**Figure 8. Different combinations of monthly relationships between a) to e) discharge, turbidity and tracers such as EC**

**and δ¹⁸O at Ponte Stelvio in 2014 and 2015. The dataset consists of n = 309 samples. Arrows underline the monthly**

**pattern.**





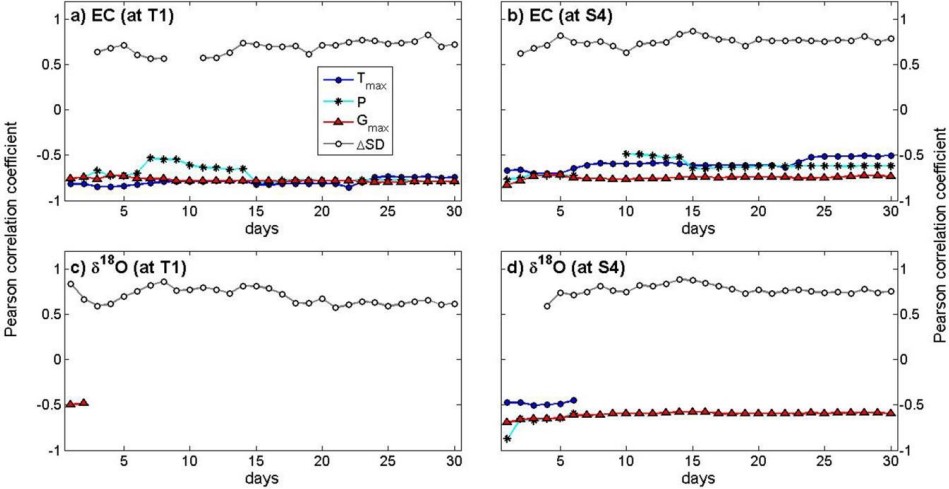

2     **Figure 9. Temporal sensitivity on the agreement of environmental variables and tracer signatures at the selected**

3     **stream locations T1 (Trafoi sub-catchment) and S4 (Sulden sub-catchment). Values are shown for a level of**

4     **significance $p < 0.05$ and missing values refer to non-significant correlations.**



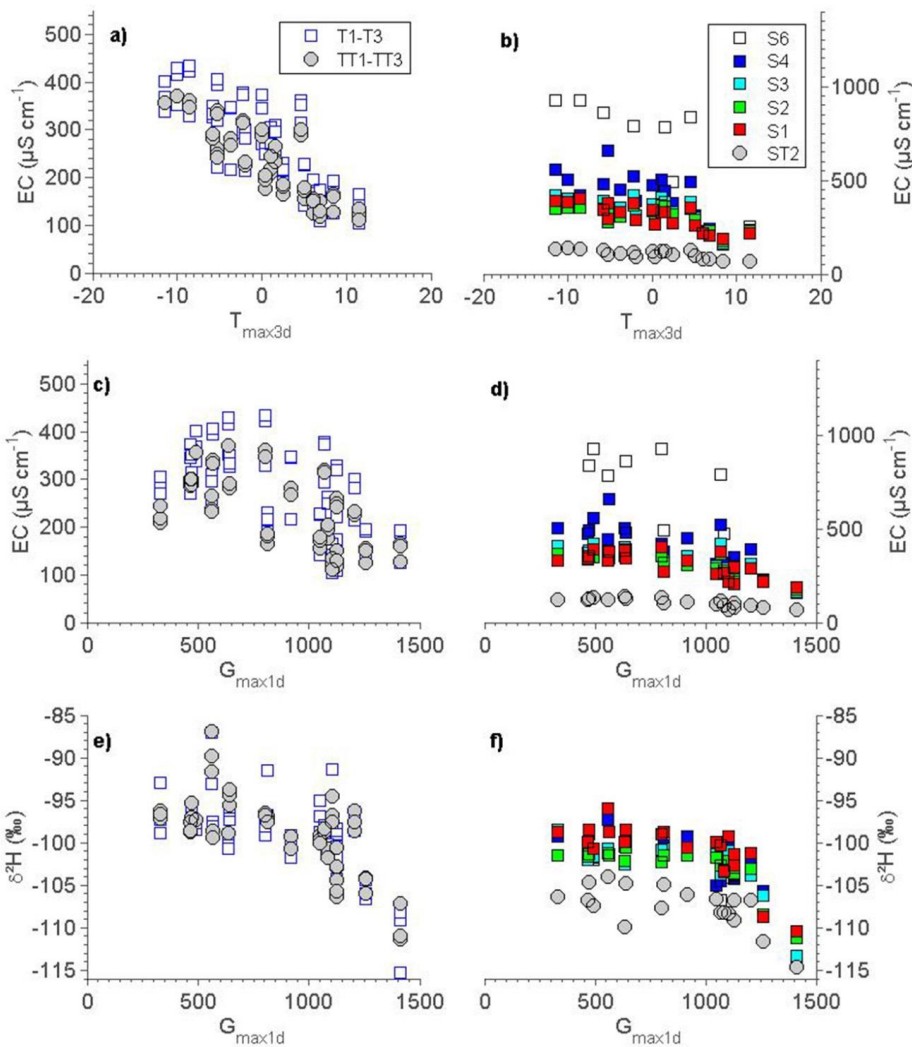

**Figure 10. Major controls of environmental variables on tracer signatures in the study area. Subplots a and b show**
**the relationship between electrical conductivity EC and daily maximum air temperature $T_{max3d}$ in the Trafoi and**
**Sulden River respectively. Subplot c and d show the relationship between electrical conductivity EC and daily**
**maximum global solar radiation $G_{max1d}$ in the Trafoi and Sulden River, respectively. Subplot e and f show the**
**relationship between $\delta^2H$ and daily maximum global radiation $G_{max1d}$ in the Trafoi and Sulden River, respectively.**





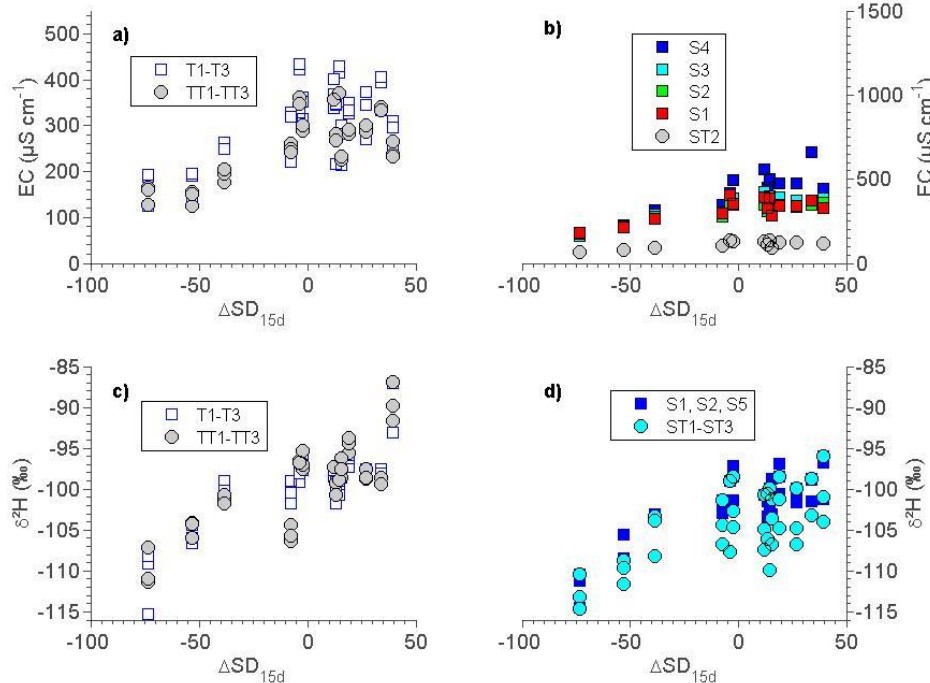

**Figure 11. Major controls of environmental variables on tracer characteristics in the study area. Subplot a and b show the relationship between electrical conductivity EC and 15 days snow depth difference $\Delta SD_{15d}$ while subplot c and d show the relationship between $\delta^2H$ and 15 days snow depth difference $\Delta SD_{15d}$ in the Trafoi and Sulden River, respectively.**