# Peer review of "Controls on spatial and temporal variability of streamflow"

_Hydrology and Earth System Sciences, 2018_

## Referee Comment (RC1) · Anonymous Referee #1 · 8 Jun 2018

The manuscript of Engel et al., explores the influences of geology and nivo-meteorological parameters on stream hydrochemistry during baseflow and during the spring/summer. They set up a comprehensive field campaign in an alpine watershed for sampling hydrochemistry. An automatic daily sample was taken from the watershed outlet; in addition monthly sampling tackled stream chemistry along two main stream branches, springs, glacial outflow, etc. By this, Engel et al., collected a high-quality spatio-temporal data set in an alpine environment. They used this for an extensive statistical data analysis and found that hydrometric and geochemical dynamics are controlled by meteorology and geological heterogeneity. The approach is derived from a previous publication of the authors in a neighboring watershed (Penna et al.,

2017), extending the analysis and suppling a novel high-quality data set. The focus on data presentation is one of the current limitations of the manuscript, already somewhat outlined in the introduction, when the authors stated that they want to fill the current research gap in alpine hydrology by "presenting data". This may be the wrong start for this research paper, as it leads to a case study and simple report style manuscript, I rather expect that a detailed question will be answered or a theory challenged by hypothesis testing. For this – the otherwise very good introduction – may need to pinpoint the research gaps more specifically and could more often summarize how something influences response rather than what influences responses of alpine watersheds (cf. lines 81 ff.). I think this will help to narrow the focus and ease the writing. This is necessary, as I feel that parts of the manuscript are somewhat premature and not fully developed yet. The current focus of the manuscript is too much on the presentation of the data and thus becomes quite a heavy read in some sections (for me), where I felt that the selection of what is important for the understanding of the research gaps and the watershed was left to the reader. A more careful selection of the data and results that are presented in detail is necessary as it will help to streamline the manuscript and better guide the reader, e.g. how far is the presentation of turbidity data relevance for the processes (among other)? The current version seems to present all derived data without carefully considering the why behind the structure of the results and the presentation thereof. This relates to the most major limitation of the work: the lack of a clear story line (already mentioned above). I found some inconsistencies in the manuscript. It starts with the title where it is stated that spatial and temporal variability of streamflow will be assessed, while neither the hypotheses, the research objectives, nor the results come back to this. Therefor I would remove the hydrometric question from the title, especially since only one station was investigated. Next, the conclusion does not clearly link back to the research objectives, and actually cover quite a range of findings from controls on streamflow chemistry to the similarity the chemical composition of glacial melt and outflow from a rock glacier; yet the main finding is rather obvious "hydrometric and geochemical dynamics were controlled by an interplay of meteorological conditions and the geological heterogeneity". Several decades have looked at this (cf. Wolock et al., 1997); the finding is very general, probably applies to nearly every watershed and one does not require such an extensive sampling campaign to be answered. A clearer analysis of the research gap, and a more specific formulation thereof (the statement about the current gaps is rather general 103-106) may help, as a very general question leads to a very general answer, and a rather speculative discussion. This is obviously ok for parts of the paper, but also a sign that the questions asked may not be specific enough for being answered with the existing data set. So I would recommend to analyze the research gap more detailed and formulate objectives/hypothesis to tackling this. From there one can tidy up the results for a better guidance (helpful to the authors and the reader). I am convinced that this is possible considering the detailed and extensive field data sets and the experience of the research group in alpine environments. Last, I felt that quality of writing declined after the introduction, you may have another careful revision before submitting the revised manuscript.

Ref:Wolock, D. M., Fan, J., & Lawrence, G. B. (1997). Effects of basin size on low-flow stream chemistry and subsurface contact time in the Neversink River watershed, New York. Hydrological Processes, 11(9), 1273-1286.

Minor comments line 68 "..., and topography with drainage ... and catchment shape" maybe delete the first "and" line 103ff, please revise and streamline. Maybe some more detail before this is needed. line 127ff, please also report mean elevation line 134, "current" does this mean 2018? Figure refers to 2006, which is not current. If 2018, why not unify this information with the figure? line 151, change "is" to "are" l155, add "At the catchment outlet" l157, the conversion to discharge is done via a rating curve, not via the salt dilution measurements. Yet, the rating curve is derived from these measurements. l164, suggestion: replace "tracer" with stream chemistry. Chemistry becomes a tracer when you infer processes, flow paths, etc. Otherwise, it is stream chemistry. l165, technically the sampling is not continuous l166, delete "Generally" l168, "respecting its seasonal variation", not sure what this should mean l172, "less than an hour"

l190ff., needs more detail l198, "before the analysis", delete "the" l223ff., "Then…" I do not understand this l232, change tracer to hydrochemical l246-256, this is a little arkward and the use of the terms old and new water quite confusing. What you actually do is calculating the discharge for the sub-catchments via the isotopic data and known discharge. So rather avoid the hydrograph separation terminology. l264, "signatures", "…area are" l312, compared to what? l329, maybe l/km2 to compare the watersheds l430ff., I found this section rather irritating. From my perception daily max temperature, max solar radiation, and the change (at least the decrease) of the snow cover are correlated. So how can you asses their impact in stream chemistry independently? Further, are you not mixing causation and correlation in this section? Hydrochemistry is caused by the amount of snow melt contributing to the streamflow, while you correlate the metrics that will lead to snowmelt with the hydrochemistry. l481ff., what is the link to the hypotheses or research question? l495ff., As connectivity is in the section header, one expects to more clearly link and discuss connectivity here, while the text itself is more about rock weathering etc. What is connected when? l569ff., see comment on l430. l594, are other met-station in the region available. Can one correlate these? The effect of topography is only marginally considered here, contrary to the sub-section's header l619ff., Why are you not performing a hydrometric data analysis? l721, this is actually something we can say about every catchment without sampling for 2 yrs l739, see comment on l430 l743ff., how? Can you show this or elaborate on this final conclusion. Table1, change "average discharge (median)" to "median discharge" Table2, can you indicate the locations in the map of Figure 1. Figure 1, add locations of table 2., font sizes are different between the subplot and too small Figure 2, please adapt the figures after fig.2 to font size and font type of this figure. Figure 4, adapt color scale of a) b) to the same range for inter-comparison, font size too small Figure 7, 9, too small Figure 10, see comment l.430

---

## Referee Comment (RC2) · Anonymous Referee #2 · 10 Jul 2018

Review for manuscript Manuscript ID: HESS-2018-135; Controls on spatial and temporal variability of streamflow in a glacierized catchment

Best authors and editors, Thank you for the possibility to review this paper, and apologies for the delay in my review. The paper studies the hydrology and hydrogeochemistry of two glaciated catchment in the Eastern Italian Alps. The work builds on a spatially and temporally distributed water sampling and monitoring campaign, supplemented with climate data. The authors are able to identify both geological and meteorological factors influencing the stream water chemistry, allowing better conceptual understanding of water sources and flow paths in the studied glaciated catchment.

I think this paper reports an impressive dataset collected in a challenging environment, and by this merit alone warrants publication. I find data analysis methods are sound, though not particularly innovative. The work has important data-based findings on the hydrology of glaciated catchments. The paper is well written with good English throughout.

general comments: I have only one major concern: with almost 3000 meters of elevation gradient and highly variable aspect and shading, only one meteorological station is used for the niveo-meteorologial variable determination. For example the snow depth (maximum depth, timing of melt) in Fig. 7 would likely be very different at different elevation ranges. The spatiotemporal variability in snowmelt at different altitudes can be a major reason for masking the tracer variability, and not creating a "coherent" tracer signal of snow and glacier melt (see discussion on L 627). Some discussion present on P20L593, but in my opinion the uncertainty caused using only one meteorological station this should be more discussed.

specific comments: P2L37: Cannot understand this sentence: what is meant with best agreement when time lengths varied?

P4L112: Why would you assume this? The hypothesis sounds somewhat trivial, and too tailored to what you found in your data.

P4L121: aim to characterize the hydrochemical signature of thawing permafrost: this does not get much attention in the rest of the manuscript, and you don't have that many water samples from permafrost thaw water either. Either reformulate the objective, or discuss the success/failure of this objective in the manuscript.

P5L141: permafrost is "sparsely located"? Can you use typical terminology for permafrost occurrence: isolated, sporadic, discontinuous.

P6L176: I'm not familiar with "rock glaciers", perhaps explain the landform when first mentioned in the text.

P8L230: do you exclude the events, where there is zero change in snow depth (no snow)? Seems so in Fig. 11.

P9L255: What do old and new water mean in this context? If I understand correctly, with Eqs 2 and 3 you are determining relative contributions from each tributary, and not any event water or other new water contribution

P9L271: I would not agree that snowmelt isotope signal is enriched from the original through the process of melting. There is an aspect of temporal variability during melting, but I would argue that the "bulk" enrichment happens through gas with water vapor exchange and sublimation in the snowpack. See e.g. Earman et al (2006) and Taylor et al (2001)

P10L284: extra parenthesis?

P11L308-321: It is not obvious when the snowmelt period is. Can you provide a hydrograph in the heat map, or describe in the text

P11L329: I don't see how the data presented shows, the relative temporal variability between the two catchment, as suggested by the authors

P12L358: discussion, not results section

P13L367: Did you measure the EC in glacier melt? Would be useful to verify the low EC water is coming from glacier melt

P14L401: wording: "clearly anticipated"?

P14L405: please indicate this event more clearly in Fig. 7, now difficult to find the data you are discussing.

P16: not sure if section 4.1 is relevant for this work. Please consider removing it, or clarify why it is important for interpreting your results.

P 17: section 4.2 is interesting speculation on the interplay between geology and hydrology, but geochemical processes discussed here goes beyond my expertise to critically evaluate the discussion.

P19L575: rephrase or remove "While $\Delta$SD was used in this study,"

P20L584: I think the control of T and G is specific to glaciated/permafrost catchments, where these variables remain important in sustaining water input even after snow has disappeared. I would not expect such a strong relationship in catchments without the possibility of thawing the glaciers/permafrost on warms days.

P20L586: I think the data you present if a bit far from providing evidence of any kind of tipping points: too speculative.

P20L612: interesting idea that the different travel times could be detectable for the correlation coefficient.

Rererences: Earman, S., A. R. Campbell, F. M. Phillips, and B. D. Newman (2006), Isotopic exchange between snow and atmospheric water vapor: Estimation of the snowmelt component of groundwater recharge in the southwestern United States, J. Geophys. Res., 111, D09302, doi:10.1029/2005JD006470.

Taylor, S., X. Feng, J. W. Kirchner, R. Osterhuber, B. Klaue, and C. E. Renshaw (2001), Isotopic evolution of a seasonal snowpack and its melt, Water Resour. Res., 37, 759–769, doi:10.1029/2000WR900341.

---

## Author Comment (AC1) · 6 Sep 2018

Response to Reviewer #1

"Controls on spatial and temporal variability of streamflow and hydrochemistry in a glacierized catchment" by Engel et al.

General comments:

The focus on data presentation is one of the current limitations of the manuscript, already somewhat outlined in the introduction, when the authors stated that they want to fill the current research gap in alpine hydrology by "presenting data". This may be

the wrong start for this research paper, as it leads to a case study and simple report style manuscript, I rather expect that a detailed question will be answered or a theory challenged by hypothesis testing. For this – the otherwise very good introduction – may need to pinpoint the research gaps more specifically and could more often summarize how something influences response rather than what influences responses of alpine watersheds (cf. lines 81 ff.). I think this will help to narrow the focus and ease the writing. This is necessary, as I feel that parts of the manuscript are somewhat premature and not fully developed yet. The current focus of the manuscript is too much on the presentation of the data and thus becomes quite a heavy read in some sections (for me), where I felt that the selection of what is important for the understanding of the research gaps and the watershed was left to the reader. A more careful selection of the data and results that are presented in detail is necessary as it will help to streamline the manuscript and better guide the reader, e.g. how far is the presentation of turbidity data relevance for the processes (among other)? The current version seems to present all derived data without carefully considering the why behind the structure of the results and the presentation thereof. This relates to the most major limitation of the work: the lack of a clear story line (already mentioned above). I found some inconsistencies in the manuscript. It starts with the title where it is stated that spatial and temporal variability of streamflow will be assessed, while neither the hypotheses, the research objectives, nor the results come back to this. Therefor I would remove the hydrometric question from the title, especially since only one station was investigated. Next, the conclusion does not clearly link back to the research objectives, and actually cover quite a range of findings from controls on streamflow chemistry to the similarity the chemical composition of glacial melt and outflow from a rock glacier; yet the main finding is rather obvious "hydrometric and geochemical dynamics were controlled by an interplay of meteorological conditions and the geological heterogeneity". Several decades have looked at this (cf. Wolock et al., 1997); the finding is very general, probably applies to nearly every watershed and one does not require such an extensive sampling campaign to be answered. A clearer analysis of the research gap, and a

more specific formulation thereof (the statement about the current gaps is rather general 103-106) may help, as a very general question leads to a very general answer, and a rather speculative discussion. This is obviously ok for parts of the paper, but also a sign that the questions asked may not be specific enough for being answered with the existing data set. So I would recommend to analyze the research gap more detailed and formulate objectives/hypothesis to tackling this. From there one can tidy up the results for a better guidance (helpful to the authors and the reader). I am convinced that this is possible considering the detailed and extensive field data sets and the experience of the research group in alpine environments. Last, I felt that quality of writing declined after the introduction, you may have another careful revision before submitting the revised manuscript.

We thank the reviewer for her/his work in reviewing this manuscript and appreciate the comments and suggestions made to guide us improving this manuscript. We share the reviewer's opinion that the story line, and thus the focus of this manuscript needs essential improvements. We will solve this aspect by better working out the research gaps and scientific contribution of such a study. With respect to the research gaps, we will focus on the following ones: • the effect of catchment characteristics and environmental conditions on stream hydrochemistry at different spatial and temporal scales. • the hydrochemical characterization of permafrost (i.e. rock glaciers as a specific form).

Furthermore, as it is closely linked to this point, we will streamline the manuscript by sharpening the research questions and providing more specific and clearer results and main messages of this work. In this context, research questions will be modified and provided as follows: 1. What is the role of geology on the hydrochemical stream signatures over time? 2. Which are the most important nivo-meteorological indicators driving stream hydrochemistry during the melting period? 3. What is the temporal relationship of discharge and tracer characteristics in the stream?

Consequently, we will also revise the manuscript regarding the order of data presented

(e.g., considering to change the order of figures) and the selection of data presented. Most likely, data referring to the LMWL are not necessarily needed to answer the research questions and therefore will be skipped.

Comment 1 line 68 ": : :, and topography with drainage and catchment shape" maybe delete the first "and"

We agree and will change it.

Comment 2 line 103ff, please revise and streamline. Maybe some more detail before this is needed.

We agree and will revise this part by evaluating carefully the research gaps.

Comment 3 line 127ff, please also report mean elevation.

We agree and will add the mean elevation of the catchment.

Comment 4 line 134, "current" does this mean 2018? Figure refers to 2006, which is not current. If 2018, why not unify this information with the figure?

We agree that this is misleading and will remove the word "current". Unfortunately, no recent data on glacier extents are available.

Comment 5 line 151, change "is" to "are" We agree and will change it.

Comment 6 l155, add "At the catchment outlet"

We agree and will change it.

Comment 7 l157, the conversion to discharge is done via a rating curve, not via the salt dilution measurements. Yet, the rating curve is derived from these measurements.

We agree and will modify as follows:"... via a flow rating curve using salt dilution/photometric measurements..."

Comment 8 l164, suggestion: replace "tracer" with stream chemistry. Chemistry be-

comes a tracer when you infer processes, flow paths, etc. Otherwise, it is stream chemistry.

We agree and will modify the subtitle to "Hydrochemical sampling and analysis".

Comment 9 l165, technically the sampling is not continuous.

We agree and will remove "continuous".

Comment 10 l166, delete "Generally"

We agree and will remove this word.

Comment 11 l168, "respecting its seasonal variation", not sure what this should mean.

We agree and will remove this part of the sentence. The 23 o'clock sampling was set to capture the early summer discharge peak, while later summer discharge peaks occur much earlier. Not knowing this at the beginning of the study, we rather preferred to be consistent with the sampling time throughout the summer.

Comment 12 l172, "less than an hour"

We agree and will change it.

Comment 13 l190ff., needs more detail

We agree that more information on snow sampling could be needed here. However, in the context of streamlining the manuscript, we will evaluate whether snow sampling data are still needed to address the research questions.

Comment 14 l198, "before the analysis", delete "the"

We agree and will change it.

Comment 15 l223ff., "Then: : :" I do not understand this

We agree and will rephrase this paragraph to make it more understandable.

Comment 16 l232, change tracer to hydrochemical

We agree and will change it.

Comment 17 l246-256, this is a little awkward and the use of the terms old and new water quite confusing. What you actually do is calculating the discharge for the sub-catchments via the isotopic data and known discharge. So rather avoid the hydrograph separation terminology.

We agree and will modify this paragraph as follows: "We will apply a two-component mixing model based on EC and ˊ2H data to separate the runoff contributions "

Comment 18 l264, "signatures",": : :area are"

We agree and will change it.

Comment 19 l312, compared to what?

We agree and will change as follows: "...reaching the most increased conductivity at S6 during the study period compared to all sampled water types,..."

Comment 20

l329, maybe l/km2 to compare the watersheds

We agree and will add a sentence on the runoff contributions translated into the specific runoff of both sub-catchments.

Comment 21 l430ff., I found this section rather irritating. From my perception daily max temperature, max solar radiation, and the change (at least the decrease) of the snow cover are correlated. So how can you asses their impact in stream chemistry independently? Further, are you not mixing causation and correlation in this section? Hydrochemistry is caused by the amount of snow melt contributing to the streamflow, while you correlate the metrics that will lead to snowmelt with the hydrochemistry.

We agree on this important comment. First, we will better describe our intention that

we aimed at providing proxy data for snowmelt, in a catchment where up to now no simulated snowmelt data are present. Second, when revising the scientific gaps and research questions, we will argue that simple nivo-meteorological indicators such as losses in snow depth being relatively easy to measure may be needed to explain changes in stream hydrochemistry. Finally, we will better separate the different parameter relationships by showing first only meteorological parameters against snow depth differences (new Figure 9) and then snow depth differences compared with discharge, EC and isotopic data (new Figure 10), to represent both hydrometric and hydrochemical stream response and avoid mixing causation and correlation. Therefore, we will remove the former Figure 9 – 11.

Comment 22 l481ff., what is the link to the hypotheses or research question?

We agree and will most likely remove this section when making the manuscript more concise.

Comment 23 l495ff., As connectivity is in the section header, one expects to more clearly link and discuss connectivity here, while the text itself is more about rock weathering etc. What is connected when?

We agree and will add few sentences or rephrase this section to focus to justify the aspect of connectivity in the subsection header.

Comment 24 l569ff., see comment on l430.

As mentioned for comment 21 (referring to l430), we will address this point when revising the corresponding result section. As suggested for comment 21, we will modify the figures so that they better reply to the research questions and avoid the issue on mixing causation and correlation.

Comment 25 l594, are other met-station in the region available. Can one correlate these? The effect of topography is only marginally considered here, contrary to the sub-section's header

We agree and will add further discussion with respect to topography. Regarding the presence of other meteorological stations, we will address it by focusing on the availability of meteorological parameter such as snow depth. We will also argue that high-elevation snowmelt (represented by snow depth differences as proxy) controls downstream isotopic stream composition due to the large amount of snow stored, being available for melting in spring.

Comment 26 l619ff., Why are you not performing a hydrometric data analysis?

We think that a hydrometric analysis is beyond the scope of this manuscript and may be addressed within future work.

Comment 27 l721, this is actually something we can say about every catchment without sampling for 2 yrs

We partly agree that this result is too vague and leads the concern addressed by the reviewer. However, as we will argue within the introduction, this kind of hydrochemical evaluation of new study sites is essential when focusing on hydrological model calibration and storages. In consequence, we will rephrase this part by providing more specific results replying to the new research questions previously posed.

Comment 28 l739, see comment on l430

We agree and will modify in accordance to our replies for comment 21 and 24.

Comment 29 l743ff., how? Can you show this or elaborate on this final conclusion.

We agree and will work on this aspect further.

Comment 30 Table1, change "average discharge (median)" to "median discharge"

We agree and will change it.

Comment 31 Table2, can you indicate the locations in the map of Figure 1.

We agree and will put the sampling locations in Figure 1.

[Figure]

Comment 32 Figure 1, add locations of table 2., font sizes are different between the subplot and too small

We agree and will modify accordingly.

Comment 33 Figure 2, please adapt the figures after fig.2 to font size and font type of this figure.

We agree and will modify accordingly. However, as stated in the reply to the reviewer's general comments, we will likely remove this figure.

Comment 34 Figure 4, adapt color scale of a) b) to the same range for inter-comparison, font size too small

With respect to Fig.5, we initially used the same range of values to better compare both sub-catchments. As Trafoi variability in EC is less pronounced than the one of Sulden, colour differences are not large enough to separate all Trafoi water sources. However, we will follow the reviewer's suggestion as the focus of this plot is more on the Trafoi – Sulden hydrochemistry comparison.

Comment 35 Figure 7, 9, too small

We agree and will modify the figure accordingly.

Comment 36 Figure 10, see comment l.430

We agree and will revise this analysis and its figures accordingly. We will make these figure more concise by, for example, focusing only on the most important nivo-meteorological indicators and changing the way these data are presented.

---

## Author Comment (AC2) · 6 Sep 2018

Response to Reviewer #2

"Controls on spatial and temporal variability of streamflow and hydrochemistry in a glacierized catchment" by Engel et al.

General comments: I have only one major concern: with almost 3000 meters of elevation gradient and highly variable aspect and shading, only one meteorological station is used for the niveo-meteorologial variable determination. For example the snow depth (maximum depth, timing of melt) in Fig. 7 would likely be very different at different el-

evation ranges. The spatiotemporal variability in snowmelt at different altitudes can be a major reason for masking the tracer variability, and not creating a "coherent" tracer signal of snow and glacier melt (see discussion on L 627). Some discussion present on P20L593, but in my opinion the uncertainty caused using only one meteorological station this should be more discussed.

We thank the reviewer for her/his work in reviewing this manuscript and appreciate the comments and suggestions made to help improving this manuscript. We agree on the concern regarding the representativeness of using data from only one meteorological station. We will address this aspect within the discussion by arguing as follows: first, the network of meteorological stations available in the study area comprises 3 high-elevation stations and 1 valley stations. However, only the Madritsch weather station as high-elevation station includes snow depth measurements. As we state in the manuscript, its elevation is similar to the lower tongue of surrounding glaciers, so that we assume its data representativeness for similar elevation bands within the catchment and thus the lower glacier covered areas. This fact motivated our aim to focus on the importance of high elevation meteorological conditions and their relation to downstream streamflow and hydrochemistry variability.

In this context, however, it is true that not only the same elevation controls snowmelt but also spatial variability such as aspect, slope, and microtopography (e.g., Anderton et al. 2002; Grünewald et al. 2010; Lopez-Moreno et al. 2013). This usually leads to different melt rates and thus affects the isotopic snowmelt signature (Taylor et al. 2001; Taylor et al. 2002; Dietermann and Weiler, 2013; Schmieder et al. 2016) and the hydrometric response in the main channel such as the timing of the discharge peak (Lundquist and Dettinger, 2005). Another point we will mention in the discussion considers the representativeness of the outlet sampling time with respect to the peak discharge time at that location. In fact, the peak of hydrochemical response may not be synchronized with the hydrometric one and therefore may lead to stronger or weaker relationships.

As a consequence of this aspect on uncertainties mentioned above and with respect also to the comment of reviewer#1 on the storyline of this manuscript, we will remove the figures 9 to 11 in its current form. Instead, we will show both the nivo-meteorological parameter variability, their relationships among each other and the temporal sensitivity of these parameters by a different graphical way, such as using boxplot diagrams. Choosing boxplots as diagram style will also underline the variability given by each parameter. Resulting from the different uncertainties associated with this data presentation, we decided that potential parameter correlations can also be derived from visual inspection.   Comment 1 P2L37: Cannot understand this sentence: what is meant with best agreement when time lengths varied?

We agree and will rephrase this part.

Comment 2 P4L112: Why would you assume this? The hypothesis sounds somewhat trivial, and too tailored to what you found in your data.

We agree and will modify the hypothesis.

Comment 3 P4L121: aim to characterize the hydrochemical signature of thawing permafrost: this does not get much attention in the rest of the manuscript, and you don't have that many water samples from permafrost thaw water either. Either reformulate the objective, or discuss the success/failure of this objective in the manuscript.

We agree that this aspect requires more care. We will consider this aspect by rephrasing the discussion accordingly. See also the response to the first comment by the first reviewer about the reformulation of the research objectives.

Comment 4 P5L141: permafrost is "sparsely located"? Can you use typical terminology for permafrost occurrence: isolated, sporadic, discontinuous.

We agree and will add "Discontinuous permafrost".

Comment 5 P6L176: I'm not familiar with "rock glaciers", perhaps explain the landform when first mentioned in the text.

We agree and will modified the sentence as follows: "As rock glaciers are considered as long term creeping ice-rock mixtures under permafrost conditions (Humlum 2000),..."

Comment 6

P8L230: do you exclude the events, where there is zero change in snow depth (no snow)? Seems so in Fig. 11.

Yes, we excluded snow depth changes between – 2 cm and + 2cm to remove noisy data. We agree that a better clarification is needed here, which we will address by adding this information.

Comment 7

P9L255: What do old and new water mean in this context? If I understand correctly, with Eqs 2 and 3 you are determining relative contributions from each tributary, and not any event water or other new water contribution

We agree and will remove the misleading sentence. A similar comment was also made by Reviewer #1.   Comment 8

P9L271: I would not agree that snowmelt isotope signal is enriched from the original through the process of melting. There is an aspect of temporal variability during melting, but I would argue that the "bulk" enrichment happens through gas with water vapour exchange and sublimation in the snowpack. See e.g. Earman et al (2006) and Taylor et al (2001)

We agree and will add these references. We will change the sentence to "...through isotopic exchange between liquid water and ice during melting conditions (Taylor et al., 2001),...".

Comment 9 P10L284: extra parenthesis?

We will remove the additional parenthesis.

Comment 10 P11L308-321: It is not obvious when the snowmelt period is. Can you provide a hydrograph in the heat map, or describe in the text

We will add more details on the melting period, to which the tracer description is referring, in the text.

Comment 11 P11L329: I don't see how the data presented shows, the relative temporal variability between the two catchment, as suggested by the authors

We agree and will address this point. The temporal description complements the spatial description of runoff contributions, previously mentioned. As these data refer to the two-component HS and are not shown in a table or figure, we will add "data not shown" at the end of this paragraph to make it clearer.

Comment 12

P12L358: discussion, not results section

We agree and will move this paragraph to the discussion section.

Comment 13 P13L367: Did you measure the EC in glacier melt? Would be useful to verify the low EC water is coming from glacier melt

Yes, we measured the EC of glacier melt and found an average EC of 36.1 $\mu$S cm-1 and an average of 13.51 ‰ in ïĄd'18O. These data and, additionally some data on snowmelt, will be reported in the text.

Comment 14 P14L401: wording: "clearly anticipated"?

We agree and will replace it by "distinctively earlier".

Comment 15 P14L405: please indicate this event more clearly in Fig. 7, now difficult to find the data you are discussing.

The period of interest is well visible from our perspective as it covers autumn 2015. However, we agree that some modifications in Fig. 7 will be helpful to improve its

visibility.

Comment 16 P16: not sure if section 4.1 is relevant for this work. Please consider removing it, or clarify why it is important for interpreting your results.

We thank for this comment and will address this aspect when restructuring the manuscript story line and its research gaps, as raised by Reviewer #1.

Comment 17 P 17: section 4.2 is interesting speculation on the interplay between geology and hydrology, but geochemical processes discussed here goes beyond my expertise to critically evaluate the discussion.

We appreciate your comment.

Comment 18 P19L575: rephrase or remove "While SD was used in this study,"

We agree and will remove this sentence.

Comment 19 P20L584: I think the control of T and G is specific to glaciated/permafrost catchments, where these variables remain important in sustaining water input even after snow has disappeared. I would not expect such a strong relationship in catchments without the possibility of thawing the glaciers/permafrost on warms days.

We agree and think that this point requires further attention. We will address it by integrating a short discussion paragraph in the discussion section.

Comment 20 P20L586: I think the data you present if a bit far from providing evidence of any kind of tipping points: too speculative.

We think that the comment on tipping points in the context of threshold-like controls is important. However, we agree that the data presented here are not exhaustive to proof the presence of general tipping point mechanism. Therefore, we will modify as follows: "...the importance of threshold-like controls at the daily and short-term scale, as described along the cascade from atmospheric circulation and local climate to hydrology...".

Comment 21 P20L612: interesting idea that the different travel times could be detectable for the correlation coefficient.

We appreciate your comment.   References:

Anderton, S., White, S. and Alvera, B.: Micro-scale spatial variability and the timing of snow melt runoff in a high mountain catchment, J. Hydrol., 268(1–4), 158–176, doi:10.1016/S0022-1694(02)00179-8, 2002.

Dietermann, N. and Weiler, M.: Spatial distribution of stable water isotopes in alpine snow cover, Hydrol. Earth Syst. Sci., 17(7), 2657–2668, doi:10.5194/hess-17-2657-2013, 2013.

Earman, S., A. R. Campbell, F. M. Phillips, and B. D. Newman: Isotopic exchange between snow and atmospheric water vapor: Estimation of the snowmelt component of groundwater recharge in the southwestern United States, J. Geophys. Res., 111, D09302, doi:10.1029/2005JD006470, 2006.

Grünewald, T., Schirmer, M., Mott, R. and Lehning, M.: Spatial and temporal variability of snow depth and ablation rates in a small mountain catchment, Cryosph., 4(2), 215–225, doi:10.5194/tc-4-215-2010, 2010.

Humlum, O.:The geomorphic significance of rock glaciers: estimates of rock glacier debris volumes and headwall recession rates in West Greenland. Geomorphology 35, 41–67, 2000.

López-Moreno, J. I., Fassnacht, S. R., Heath, J. T., Musselman, K. N., Revuelto, J., Latron, J., Morán-Tejeda, E. and Jonas, T.: Small scale spatial variability of snow density and depth over complex alpine terrain: Implications for estimating snow water equivalent, Adv. Water Resour., 55, 40–52, doi:10.1016/j.advwatres.2012.08.010, 2013.

Lundquist, J. D. and Dettinger, M. D.: How snowpack heterogeneity affects diurnal streamflow timing, Water Resour. Res., 41, 1–14, doi:10.1029/2004WR003649, 2005.

Schmieder, J., Hanzer, F., Marke, T., Garvelmann, J., Warscher, M., Kunstmann, H., and Strasser, U.: The importance of snowmelt spatiotemporal variability for isotope-based hydrograph separation in a high-elevation catchment, Hydrol. Earth Syst. Sci., 20, 5015-5033, https://doi.org/10.5194/hess-20-5015-2016, 2016.

Taylor, S., Feng, X., Kirchner, J. W., Osterhuber, R., Klaue, B. and Renshaw, C. E.: Isotopic evolution of a seasonal snowpack and its melt, Water Resour. Res., 37(3), 759–769, 2001.

Taylor, S., Feng, X., Williams, M. and McNamara, J.: How isotopic fractionation of snowmelt affects hydrograph separation, Hydrol. Process., 16(18), 3683–3690, doi:10.1002/hyp.1232, 2002.
* * *

---

## Author Response (AR1)

[revised manuscript text omitted]

General comments:

The focus on data presentation is one of the current limitations of the manuscript, already somewhat outlined in the introduction, when the authors stated that they want to fill the current research gap in alpine hydrology by "presenting data". This may be the wrong start for this research paper, as it leads to a case study and simple report style manuscript, I rather expect that a detailed question will be answered or a theory challenged by hypothesis testing. For this – the otherwise very good introduction – may need to pinpoint the research gaps more specifically and could more often summarize how something influences response rather than what influences responses of alpine watersheds (cf. lines 81 ff.). I think this will help to narrow the focus and ease the writing. This is necessary, as I feel that parts of the manuscript are somewhat premature and not fully developed yet. The current focus of the manuscript is too much on the presentation of the data and thus becomes quite a heavy read in some sections (for me), where I felt that the selection of what is important for the understanding of the research gaps and the watershed was left to the reader. A more careful selection of the data and results that are presented in detail is necessary as it will help to streamline the manuscript and better guide the reader, e.g. how far is the presentation of turbidity data relevance for the processes (among other)? The current version seems to present all derived data without carefully considering the why behind the structure of the results and the presentation thereof. This relates to the most major limitation of the work: the lack of a clear story line (already mentioned above). I found some inconsistencies in the manuscript. It starts with the title where it is stated that spatial and temporal variability of streamflow will be assessed, while neither the hypotheses, the research objectives, nor the results come back to this. Therefor I would remove the hydrometric question from the title, especially since only one station was investigated. Next, the conclusion does not clearly link back to the research objectives, and actually cover quite a range of findings from controls on streamflow chemistry to the similarity the chemical composition of glacial melt and outflow from a rock glacier; yet the main finding is rather obvious "hydrometric and geochemical dynamics were controlled by an interplay of meteorological conditions and the geological heterogeneity". Several decades have looked at this (cf. Wolock et al., 1997); the finding is very general, probably applies to nearly every watershed and one does not require such an extensive sampling campaign to be answered. A clearer analysis of the research gap, and a more specific formulation thereof (the statement about the current gaps is rather general 103-106) may help, as a very general question leads to a very general answer, and a rather speculative discussion. This is obviously ok for parts of the paper, but also a sign that the questions asked may not be specific enough for being answered with the existing data set. So I would recommend to analyze the research gap more detailed and formulate objectives/hypothesis to tackling this. From there one can tidy up the results for a better guidance (helpful to the authors and the reader). I am convinced that this is possible considering the detailed and extensive field data sets and the experience of the research group in alpine environments. Last, I felt that quality of writing declined after the introduction, you may have another careful revision before submitting the revised manuscript.

We thank the reviewer for her/his work in reviewing this manuscript and appreciate the comments and suggestions made to guide us improving this manuscript.

We share the reviewer's opinion that the story line, and thus the focus of this manuscript needed essential improvements. We solved this aspect by better working out the research gaps and scientific contribution of such a study.

With respect to the research gaps, we focused on the following ones:

- the effect of catchment characteristics and environmental conditions on stream hydrochemistry at different spatial and temporal scales.
- the hydrochemical characterization of permafrost (i.e. rock glaciers as a specific form).

Furthermore, as it is closely linked to this point, we streamlined the manuscript by sharpening the research questions and providing more specific and clearer results and main messages of this work.

In this context, research questions were modified and provided as follows:

1. What is the role of geology on the hydrochemical stream signatures over time?
2. Which are the most important nivo-meteorological indicators driving stream hydrochemistry during the melting period?
3. What is the temporal relationship of discharge and tracer characteristics in the stream?

Consequently, we also revised the manuscript regarding the order of data presented (e.g., considering to change the order of figures) and the selection of data presented. Data referring to the LMWL were not necessarily needed to answer the research questions and therefore were be skipped.

**Comment 1**
line 68 ": : :, and topography with drainage and catchment shape" maybe delete the first "and"

**We agree and changed it.**

**Comment 2**
line 103ff, please revise and streamline. Maybe some more detail before this is needed.

**We agree and revised this part by evaluating carefully the research gaps.**

**Comment 3**
line 127ff, please also report mean elevation.

**We agree and added the mean elevation of the catchment.**

**Comment 4**
line 134, "current" does this mean 2018? Figure refers to 2006, which is not current. If 2018, why not unify this information with the figure?

**We agree that this is misleading and removed the word „current". Unfortunately, no recent data on glacier extents are available.**

**Comment 5**
line 151, change "is" to "are"

**We agree and changed it.**

**Comment 6**
l155, add "At the catchment outlet"

**We agree and changed it.**

**Comment 7**
l157, the conversion to discharge is done via a rating curve, not via the salt dilution measurements. Yet, the rating curve is derived from these measurements.

**We agree and modified as follows:"... via a flow rating curve using salt dilution/photometric measurements…"**

**Comment 8**
l164, suggestion: replace "tracer" with stream chemistry. Chemistry becomes a tracer when you infer processes, flow paths, etc. Otherwise, it is stream chemistry.

**We agree and modified the subtitle to „Hydrochemical sampling and analysis".**

**Comment 9**
l165, technically the sampling is not continuous.

**We agree and removed "continuous".**

**Comment 10**
l166, delete "Generally"

**We agree and removed this word.**

**Comment 11**
l168, "respecting its seasonal variation", not sure what this should mean.

**We agree and removed this part of the sentence. The 23 o'clock sampling was set to capture the early summer discharge peak, while later summer discharge peaks occur much earlier. Not knowing this at the beginning of the study, we rather preferred to be consistent with the sampling time throughout the summer.**

**Comment 12**
l172, "less than an hour"

**We agree and changed it.**

**Comment 13**
l190ff., needs more detail

We agree that more information on snow sampling could be needed here. However, in the context of streamlining the manuscript, we decided that further snow sampling data was not needed to address the research questions.

**Comment 14**
l198, "before the analysis", delete "the"

**We agree and changed it.**

**Comment 15**
l223ff., "Then: : :" I do not understand this

**We agree and rephrased this paragraph to make it more understandable.**

**Comment 16**
l232, change tracer to hydrochemical

**We agree and changed it.**

**Comment 17**
l246-256, this is a little awkward and the use of the terms old and new water quite confusing. What you actually do is calculating the discharge for the sub-catchments via the isotopic data and known discharge. So rather avoid the hydrograph separation terminology.

**We agree and modified this paragraph as follows: "We applied a two-component mixing model based on EC and $\delta^2$H data to separate the runoff contributions originating from the Sulden and Trafoi sub-catchment at each sampling moment during monthly sampling "**

**Comment 18**
l264, "signatures",": : :area are"

**We agree and changed it.**

**Comment 19**
l312, compared to what?

**We agree and changed as follows: "...reaching the most increased conductivity at S6 during the study period compared to all sampled water types,…"**

**Comment 20**

l329, maybe l/km2 to compare the watersheds

**We agree and added a sentence on the runoff contributions translated into the specific runoff of both sub-catchments.**

**Comment 21**

l430ff., I found this section rather irritating. From my perception daily max temperature, max solar radiation, and the change (at least the decrease) of the snow cover are correlated. So how can you asses their impact in stream chemistry independently? Further, are you not mixing causation and correlation in this section? Hydrochemistry is caused by the amount of snow melt contributing to the streamflow, while you correlate the metrics that will lead to snowmelt with the hydrochemistry.

**We agree on this important comment. First, we better described our intention that we aimed at providing proxy data for snowmelt, in a catchment where up to now no simulated snowmelt data are present. Second, when revising the scientific gaps and research questions, we decided that simple nivo-meteorological indicators such as losses in snow depth being relatively easy to measure may be needed to explain changes in stream hydrochemistry. Finally, we better separated the different parameter relationships by showing first only meteorological parameters against snow depth differences (new Figure 6) and then snow depth differences compared with discharge, EC and isotopic data (new Figure 7), to represent both hydrometric and hydrochemical stream response and avoid mixing causation and correlation.**

**Comment 22**
l481ff., what is the link to the hypotheses or research question?

**We agree and removed this section when making the manuscript more concise.**

**Comment 23**
l495ff., As connectivity is in the section header, one expects to more clearly link and discuss connectivity here, while the text itself is more about rock weathering etc. What is connected when?

**We agree and removed "connectivity" from the subsection header.**

**Comment 24**
l569ff., see comment on l430.

**As mentioned for comment 21 (referring to l430), we addressed this point when revising the corresponding result section. As suggested for comment 21, we modified the figures so that they better reply to the research questions and avoid the issue on mixing causation and correlation.**

**Comment 25**
l594, are other met-station in the region available. Can one correlate these? The effect of topography is only marginally considered here, contrary to the sub-section's header

**We agree and removed "typography" for that reason. Regarding the presence of other meteorological stations, we addressed it by focusing on the availability of meteorological parameter such as snow depth. We also argued that high-elevation snowmelt (represented by snow depth differences as proxy) controls downstream isotopic stream composition due to the large amount of snow stored, being available for melting in spring.**

**Comment 26**

l619ff., Why are you not performing a hydrometric data analysis?

**We think that a hydrometric analysis is beyond the scope of this manuscript. However, such as analysis may be addressed within future work.**

**Comment 27**

l721, this is actually something we can say about every catchment without sampling for 2 yrs

**We partly agree that this result is too vague and leads the concern addressed by the reviewer. However, as we will argue within the introduction, this kind of hydrochemical evaluation of new study sites is essential when focusing on hydrological model calibration and storages. In consequence, we will rephrase this part by providing more specific results replying to the new research questions previously posed.**

**Comment 28**

l739, see comment on l430

**We agree and modified in accordance to our replies for comment 21 and 24.**

**Comment 29**

l743ff., how? Can you show this or elaborate on this final conclusion.

**We agree that this could be an important aspect to study on in future.**

**Comment 30**

Table1, change "average discharge (median)" to "median discharge"

**We agree and changed it.**

**Comment 31**

Table2, can you indicate the locations in the map of Figure 1.

**The sampling locations were already present in the map. However, for better readability, we put the labels of each sampling locations.**

**Comment 32**

Figure 1, add locations of table 2., font sizes are different between the subplot and too small

**We agree and modified accordingly.**

**Comment 33**

Figure 2, please adapt the figures after fig.2 to font size and font type of this figure.

**We agree on this comment. However, as stated in the reply to the reviewer's general comments, we decided to remove this figure for streamlining the paper.**

**Comment 34**

Figure 4, adapt color scale of a) b) to the same range for inter-comparison, font size too small

**With respect to Fig.5, we initially used the same range of values to better compare both sub-catchments. As Trafoi variability in EC is less pronounced than the one of Sulden, colour differences are not large enough to separate all Trafoi water sources. However, we followed the reviewer's suggestion as the focus of this plot is more on the Trafoi – Sulden hydrochemistry comparison.**

**Comment 35**
Figure 7, 9, too small

**We agree and modified the figure accordingly.**

**Comment 36**
Figure 10, see comment l.430

**We agree and revised this analysis and its figures accordingly. We produced figures on the most important nivo-meteorological indicators (air temperature and global solar radiator) related to variability of snow depth losses and the latter one compared with discharge, EC and $\delta^{18}$O.**

**Response to Reviewer #2**

**"Controls on spatial and temporal variability of streamflow and hydrochemistry in a glacierized catchment" by Engel et al.**

General comments:
I have only one major concern: with almost 3000 meters of elevation gradient and highly variable aspect and shading, only one meteorological station is used for the niveo-meteorologial variable determination. For example the snow depth (maximum depth, timing of melt) in Fig. 7 would likely be very different at different elevation ranges. The spatiotemporal variability in snowmelt at different altitudes can be a major reason for masking the tracer variability, and not creating a "coherent" tracer signal of snow and glacier melt (see discussion on L 627). Some discussion present on P20L593, but in my opinion the uncertainty caused using only one meteorological station this should be more discussed.

**We thank the reviewer for her/his work in reviewing this manuscript and appreciate the comments and suggestions made to help improving this manuscript. We agree on the concern regarding the representativeness of using data from only one meteorological station. We addressed this aspect within the discussion by arguing as follows: first, the network of meteorological stations available in the study area comprises 3 high-elevation stations and 1 valley stations. However, only the Madritsch weather station as high-elevation station includes snow depth measurements. As we stated in the manuscript, its elevation is similar to the lower tongue of surrounding glaciers, so that we assume its data representativeness for similar elevation bands within the catchment and thus the lower glacier covered areas. This fact motivated our aim to focus on the importance of high elevation meteorological conditions and their relation to downstream streamflow and hydrochemistry variability.**

**In this context, however, it is true that not only the same elevation controls snowmelt but also spatial variability such as aspect, slope, and microtopography (e.g., Anderton et al. 2002; Grünewald et al. 2010; Lopez-Moreno et al. 2013). This usually leads to different melt rates and thus affects the isotopic snowmelt signature (Taylor et al. 2001; Taylor et al. 2002; Dietermann and Weiler, 2013; Schmieder et al. 2016) and the hydrometric response in the main channel such as the timing of the discharge peak (Lundquist and Dettinger, 2005). Another point we will mention in the discussion considers the representativeness of the outlet sampling time with respect to the peak discharge time at that location. In fact, the peak of hydrochemical response may not be synchronized with the hydrometric one and therefore may lead to stronger or weaker relationships.**

**As a consequence of this aspect on uncertainties mentioned above and with respect also to the comment of reviewer#1 on the storyline of this manuscript, we removed the former figures 9 to 11 in its current form. Instead, we showed both the nivo-meteorological parameter variability, their relationships among each other and the temporal sensitivity of these parameters by using box-plots diagrams. Choosing boxplots as diagram style also underlined the variability given by each parameter. Resulting from the different uncertainties associated with this data presentation, we decided that potential parameter correlations can also be derived from visual inspection.**

**Comment 1**

P2L37: Cannot understand this sentence: what is meant with best agreement when time lengths varied?

**We agree and rephrased this part.**

**Comment 2**

P4L112: Why would you assume this? The hypothesis sounds somewhat trivial, and too tailored to what you found in your data.

**We agree and modified the hypothesis in a first step. During further revision, however, we removed the hypothesis but reworked the specific aims.**

**Comment 3**

P4L121: aim to characterize the hydrochemical signature of thawing permafrost: this does not get much attention in the rest of the manuscript, and you don't have that many water samples from permafrost thaw water either. Either reformulate the objective, or discuss the success/failure of this objective in the manuscript.

**We agree that this aspect requires more care. We considered this aspect by reformulating the research objectives. See also the response to the first comment by the first reviewer about the reformulation of the research objectives.**

**Comment 4**

P5L141: permafrost is "sparsely located"? Can you use typical terminology for permafrost occurrence: isolated, sporadic, discontinuous.

**We agree and added "Permafrost is discontinuously located…".**

**Comment 5**

P6L176: I'm not familiar with "rock glaciers", perhaps explain the landform when first mentioned in the text.

**We agree and modified the sentence as follows: "Three outflows from two active rock glaciers were selected to represent meltwater from permafrost because rock glaciers are considered as long term creeping ice-rock mixtures under permafrost conditions (Humlum 2000)"**

**Comment 6**

P8L230: do you exclude the events, where there is zero change in snow depth (no snow)? Seems so in Fig. 11.

**Yes, we excluded snow depth changes between – 2 cm and + 2cm to remove noisy data. However, in the new manuscript version, we wrote "Then, we excluded snow depth losses up to 5 cm to remove noisy data" as we modified data analysis during reviewing.**

**Comment 7**

P9L255: What do old and new water mean in this context? If I understand correctly, with Eqs 2 and 3 you are determining relative contributions from each tributary, and not any event water or other new water contribution

**We agree and removed the misleading sentence. A similar comment was also made by Reviewer #1.**

**Comment 8**

P9L271: I would not agree that snowmelt isotope signal is enriched from the original through the process of melting. There is an aspect of temporal variability during melting, but I would argue that the "bulk" enrichment happens through gas with water vapour exchange and sublimation in the snowpack. See e.g. Earman et al (2006) and Taylor et al (2001)

**We agree and added these references. In a previous version, we changed the sentence to "…through isotopic exchange between liquid water and ice during melting conditions (Taylor et al., 2001),…". However, we skipped this paragraph during the streamlining process.**

**Comment 9**

P10L284: extra parenthesis?

**We removed the additional parenthesis.**

**Comment 10**

P11L308-321: It is not obvious when the snowmelt period is. Can you provide a hydrograph in the heat map, or describe in the text

**We think that more details on the melting period are not needed here. However, the hydrograph of the outlet is shown in Fig. 8 with the labels of the sampling day, which are first introduced in Fig. 4.**

**Comment 11**

P11L329: I don't see how the data presented shows, the relative temporal variability between the two catchment, as suggested by the authors

**We agree and addressed this point. The temporal description complements the spatial description of runoff contributions, previously mentioned. We added a sentence on the runoff contributions translated into the specific runoff of both sub-catchments (see comment 20 for Reviewer #1)**

**Comment 12**

P12L358: discussion, not results section

**We agree and removed this paragraph.**

**Comment 13**

P13L367: Did you measure the EC in glacier melt? Would be useful to verify the low EC water is coming from glacier melt

**Yes, we measured the EC of glacier melt and found an average EC of 36.1 µS cm-1 and an average of 13.51 ‰ in $\delta^{18}$O. These data and, additionally some data on snowmelt, are reported now in the text.**

**Comment 14**

P14L401: wording: "clearly anticipated"?

**We agree and replaced it by "distinctively earlier".**

**Comment 15**

P14L405: please indicate this event more clearly in Fig. 7, now difficult to find the data you are discussing.

**The period of interest is well visible from our perspective as it covers autumn 2015.**

**Comment 16**
P16: not sure if section 4.1 is relevant for this work. Please consider removing it, or clarify why it is important for interpreting your results.

**We thank for this comment and addressed this aspect when restructuring the manuscript story line and its research gaps, as raised by Reviewer #1.**

**Comment 17**
P 17: section 4.2 is interesting speculation on the interplay between geology and hydrology, but geochemical processes discussed here goes beyond my expertise to critically evaluate the discussion.

**We appreciated your comment.**

**Comment 18**
P19L575: rephrase or remove "While SD was used in this study,"

**We agree and removed this sentence.**

**Comment 19**
P20L584: I think the control of T and G is specific to glaciated/permafrost catchments, where these variables remain important in sustaining water input even after snow has disappeared. I would not expect such a strong relationship in catchments without the possibility of thawing the glaciers/permafrost on warms days.

**We agree and think that this point requires further attention. We addressed it by adding few setences on threshold-like behaviour of these meteorological indicators in the discussion section 5.2.**

**Comment 20**
P20L586: I think the data you present if a bit far from providing evidence of any kind of tipping points: too speculative.

**We think that the comment on tipping points in the context of threshold-like controls is important. However, we agree that the data presented here are not exhaustive to proof the presence of general tipping point mechanism. Therefore, we removed this paragraph from the manuscript.**

**Comment 21**
P20L612: interesting idea that the different travel times could be detectable for the correlation coefficient.

**We appreciated your comment.**

---

## Author Response (AR2)

**Response to Reviewer #2**

**"Controls on spatial and temporal variability of streamflow and hydrochemistry in a glacierized catchment" by Engel et al.**

**General comments:**

Engel et al. use hydrochemistry to assess what the most important runoff-generation mechanisms in glacierized catchments are, and how they change temporally and spatially as a function of geology and climate. They do so by presenting an extensive field campaign performed over the course of two years (2014 and 2015) to sample stable isotopes, electrical conductivity, and element concentration. They interpret these results by leveraging additional streamflow and turbidity data as well as snow, and weather data from one ground-based station at high elevation.

 Overall, the research presented here is relevant and in line with the audience and the interests of HESS. That said, I think that the framing, the presentation, and the discussion of these results would need some improvements before publication. Because of all the points I will detail below, I suggest the editor reconsider this manuscript after a minor-revision round. In fact, while the amount of comments is rather substantial, they mostly regard text clarity and do not regard the analyses, which I found quite robust.

A first point that I found quite confusing while reading this manuscript is the lack (at least to me) of an evident research hypothesis that could justify the research methods, frame the choice of the study area in light of the international literature, and thus make the contribution and the message of this manuscript specific enough to fit into one single scientific paper. In the introduction, authors say that "although the effect of catchment characteristics and environmental conditions on stream hydrochemistry at different spatial and temporal scales has well been studied in lowland and mid-land catchments (e.g. Wolock et al., 1997; McGuire et al. 2005; Tetzlaff et al., 2009), only few studies have focused on this aspect in glacierized or permafrost-dominated catchments". In my understanding, this should be the main knowledge gap that Engel et al. have tried to fill, but at this stage it still reads quite broadly. Thus, they conclude the Introduction by formulating three research questions (see lines 121 – 125).

Of course, understanding the role of geology on the hydrochemical stream signatures over time is relevant and timely, as it is to clarify which are the most important nivo-meteorological indicators driving stream hydrochemistry during the melting period. However, each of these goals is so broad that could be the target of several stand-alone papers. The consequence of setting such diverse goals is that results tend (at least to me) to become a bit dispersive and general (see e.g. the first two lines of the conclusions). A second consequence is that readers (especially the international ones) lack the framework needed to understand, among others, why this extensive campaign in these specific catchments could contribute to global hydrology and hydrochemistry. So my first major suggestion is that authors could (1) choose one of the three research domains currently introduced at lines 121-125; (2) state and comment the specific hypotheses they would like to test and thus the reasons that led to the choice of this study area; (3) reframe the introduction and the remainder of the paper according to the main key findings with regard to these specific hypotheses; (4) elaborate on the discussion section to expand the implications of this work for other regions of the world where similar studies would be beneficial.

**We thank for these comments and understand the concerns expressed by the reviewer. We think that the storyline became much clearer compared to previous versions of the manuscript because we better addressed the research gap and modified the specific aims. We argue that the complexity of the hydrochemical responses observed in our study catchment and the analysis of their controls deserve such a broad, and partly qualitative perspective. However, we agree with this comment and modified the research gap paragraph and re-defined the specific objectives.**

**We changed the last paragraph of the introduction to better underline the research gaps, leading to the main research hypothesis: "We hypothesise that the markedly different geological properties affect the geochemistry and the hydrological response of both catchments."**

**Consequently, we modified two of the specific objectives as follows:**

**• Does the temporal pattern of the hydrochemical stream signature in the two catchments reflect the dominant rock substratum?**

**• Do nivo-meteorological indicators (precipitation, air temperature, solar radiation, snow depth) impact the stream hydrochemical response during the melting period?**

**Moreover, we inserted the following sentence in the discussion section 4.3: "This aspect finally underlines the need for conducting multi-tracer studies in glacierized catchments with different geological complexity, in order to evaluate whether our findings (obtained in sedimentary and metamorphic substratum) are transferable to different geological settings."**

I also found the language of the manuscript sometimes too qualitative. My (personal) opinion is that this is again due – at least partially – to the breadth of the research questions that the manuscript is trying to answer. A few examples, with my comments in square brackets:

"Results highlight the dominant impact of water enriched in solutes during baseflow conditions starting from late autumn to early spring prior to the onset of the melting period in May/June of both years [is there any way you can replace this "dominant role" with something more quantitative and specific? This also sounds more like a discussion sentence and I would expect results to focus on metrics that could quantify this impact rather than saying that "they highlight the dominant impact"]. Such an impact seemed to be highest [how did you measure this? Consider replacing "seemed" with something more definitive] in water from streams and tributaries reaching the most increased conductivity [I would explicitly mention numbers here rather than "the most increased"] at S6 during the study period compared to all sampled water types, ranging from 967 to 992 µS cm-1 in January to March 2015. During the same period of time, isotopic composition was slightly more enriched [how much?] and spatially more homogeneous [how much?] among the stream, tributaries, and springs than in the summer months." (lines 294 – 300);

**We understand this comment. We have thus reported more isotopic data in the text that describes the results to be more quantitative. Furthermore, we shortened this paragraph to facilitate reading.**

"In contrast, the Sulden River revealed relatively high EC [how much? Relatively to what benchmark?] at the highest upstream location (S6) and relatively low EC upstream [same as before] the confluence with the Trafoi River (S2) during baseflow conditions. The exponential decrease in EC ('EC dilution gradient') during this period of time was strongly linked to the catchment area [how did you measure this strong link?]" (lines 326 - 329) ;

**We now provide the relevant EC data for S6 and S2. We also added the coefficient of correlation (already displayed in Fig. 5) to underline the strong exponential link between dilution gradient and catchment area.**

"Furthermore, the interannual variability of meteorological conditions with respect to the occurrence of warm days [warm is relative, I would replace with numbers – maybe days with avg temperature greater than 0 degC or 5 degC as done for the snow-melt analysis?], storm events and snow cover of the contrasting years 2014 and 2015 is clearly visible [this is also relative, consider measuring with some metrics] and contributed to the hydrochemical dynamics (Fig.8 and Table 1). (…) In contrast, warmer days in 2014 were less pronounced and frequent [provide statistics] but accompanied by intense storms of up to 50 mm d-1. These meteorological conditions seem to contribute [I would replace this with something more definitive and informative] to the general hydrochemical patterns described above. (lines 403 – 417).

**Regarding the comment on warm days, we added the number of days when the daily maximum air temperature exceeded 6.5°C (the entire catchment is above freezing conditions) and 15°C to represent heat waves. Furthermore, we rephrased the second sentence mentioned in the reviewer's comment and removed the qualitative description. We also report the number of days when intense storms for up to 50 mm d$^{-1}$ occurred. We finally removed the last sentence "These meteorological conditions seem to contribute…".**

SPECIFIC COMMENTS (each of these comments start with the line number to which it refers)

**Comment 1**

- 60: what does "this objective" refer to here?

**We modified this sentence as follows "this objective" referred to the previous paragraph, where the understanding of catchment behaviour is mentioned.**

**Comment 2**

- 65-66: what do you mean with "topography with drainage network"?

**We modified this sentence. The drainage network is not necessarily linked to topography, although the sentences structure might have implied it.**

**Comment 3**

- 74: maybe replace "address" with "quantify" or "clarify"?

**We changed it.**

**Comment 4**

- 77: to me, streamflow would (at least partially) correlate with air temperature even in other circumstances, e.g., Mediterranean catchments were temperature-driven ET is an important driver of water supply.

**Yes, we agree that ET might become an important control of water supply. However, this usually holds for Mediterranean climate. In mountainous catchments, a previous study from the Swiss Alps showed the importance of ET controlling the hydrometric response (Mutzner et al., 2017).**

**Comment 5**

- 115: it was difficult to me to understand what is the specific "gap" that you aim to address here. If this is what is reported at lines 101-106, then the paragraph about permafrost makes the link misleading as it breaks the flow of information.

**We agree and moved the corresponding paragraph to the previous section, where permafrost is already addressed.**

**Comment 6**

- 115: two year -> two-year

**We changed it.**

**Comment 7**

- 136: how does this glacier area in 2006 compare with more recent estimates from, e.g., the Randolph Glacier Inventory v6 released in July 2017?

**This is an important comment. In fact, the Randolph Glacier Inventory v6 contains glacier extents for 2011 and 2017. As the latter data would refer to a period not being addressed in this study, we decided to insert the 2011 data.**

**Therefore, we replaced the 2006 extent with the 2011 extent. Furthermore, we changed the corresponding glacier proportion for each catchment in Table II and displayed it in Fig. 1c (Smiraglia, 2015).**

**Comment 8**

- 140-143: it would be helpful to include more information on geological properties that are intuitive for a general audience, such as permeability or percentage of clay and silt in the soil layer (if at all available). These properties would help to relate these geological groups with expected infiltration patterns and thus runoff response.

**Unfortunately, data on rock permeability and the extent and composition of clay in the study basin are not available.**

**Comment 9**

- 158: maybe this was already discussed during the first round of revision, but could you comment on the expected representativeness of this station for both sub-basins?

**We agree on this comment, which indeed was already pointed out in the previous revision. We addressed this aspect within the discussion by arguing as follows: first, the network of meteorological stations available in the study area comprises 3 high-elevation stations and 1 valley station. However, only the Madritsch weather station – a high-elevation station - includes snow depth measurements. As we stated in the manuscript, its elevation is very close to that of the surrounding glaciers tongues, so that we can assume its representativeness these areas.**

**Comment 10**

- 178: could you quantify what do you mean with "very limited"? That is, could you provide statistics to make this more informative?

**We argue that the hydrochemical variability at the daily scale during winter baseflow conditions is neglectable as also the discharge is constant due to the lack of melt water inputs. We support this by adding a new reference (Immerzeel et al., 2012). As electrical conductivity meters did not function during winter months, only discharge measurements at the outlet were available to show that no daily discharge variation occurred during baseflow conditions (1.1 – 3.5 m³ s$^{-1}$) compared to discharge variability during melt period (4.4 – 80.8 m³ s$^{-1}$).**

**Comment 11**

- 228ff: assuming snow-depth decreases as a proxy of snow melt is a simplified approach, but authors are clear on this point (see also the discussion section). As a side note, I would suggest authors convert the Delta SD data reported throughout the manuscript (e.g., see Figure 6) into snow-melt runoff, which can be estimated from Delta SD via an assumption on snow density. The advantage is that, in the snow-hydrology literature, snow-melt runoff is usually assumed positive as it is an input to the stream network (the larger, the more snow has melted). This would make result interpretation a bit easier to follow (e.g., snow-melt runoff would increase with radiation or air temperature in Fig. 6 as a diagonal reader would expect).

**With respect to melt dynamics and related controls, we agree that measured snowmelt data would be much more appropriate to compare with the environmental indicators. However, from other field data (not used in this study) ,we know that a simple snow density assumption to infer SWE is prone to errors as snow density strongly depends on the snow layers and is highly variable within the snow pack both in space (due to elevation, aspect and micro-topography) and over time (due to seasonality) increasing in spring. So, we think that converting ΔHS in ΔSWE without a proper snow model would introduce further uncertainty in the analysis.**

**Comment 12**

- 245: I may have missed how baseflow and melt periods were defined.

**We added the period of time we refer to in Line 175 and line 183.**

**Comment 13**

- 256: remove "the" before "this"

**We corrected it.**

**Comment 14**

- Tables 4 and 5 are quite challenging to screen and understand, especially for diagonal readers. What about replacing them with something like a boxplot of VC where heavy metals and other elements are depicted with different colors, and move these tables to a supplement? Sounds like the main point here is the difference in chemical composition during snow-melt and baseflow, something that VC should easily measure (and indeed, VC is the main metric used to make this point in this section). I also found difficult to understand how "The observed geochemical patterns are confirmed by PCA results (Fig. 2) and the correlation matrix (Fig. 3)" – maybe a few words on this could be helpful.

**Thank for this suggestion. We decided to move Table 4 and 5 to the supplementary material but keeping the data as they are. Boxplots might be visually nicer but we prefer to show the values, which are easier to depict in this way of representation. We underline again that VC is inferred from the variability of SD during baseflow and melt period, instead of displaying single values for each sampling day and each element concentration.**

**The geochemical patterns (1. high heavy metal concentration during melting period; 2. increase of As, Sr, K, Sb during baseflow conditions) is mentioned in the previous lines and can easily be seen from Fig. 2. The text added to the figure should help the reader as well to better identify the hydrological meaning of the axis.**

 **Comment 15**

 - 363: passed -> exceeded

**We corrected it.**

**Comment 16**

 - Fig. 6 vs. 7: in Fig. 7, the range of snow-depth differences spans -150 cm and -50, but in Fig. 6 the minimum difference is about -80 cm. Am I missing something here?

**Thanks, there was indeed a labelling mistake on the x axis, which we corrected. The snow depth data, on which Figure 6 and 7 are based, are the same.**

**Comment 17**

 - Fig. 8: the color of the line for turbidity is not clear to me

**The brownish line in Fig.8 c and d refers to turbidity. The symbol in the legend might be too thin to be visible. Therefore, we slightly enlarged them, but we wanted to avoid overlapping with the discharge timeseries.**

**Comment 18**

- 402: what flood are you referring to here?

**We describe here the turbidity values of a flood event occurring in mid-August 2014 (see Line 405). The sentences is as follows: "the maximum value recorded was 1904 NTU reached after several storm events of different precipitation amounts (17 mm, 50 mm, and 9 mm) on 12, 13, and 14 August 2014, respectively."**

**Comment 19**

- 410: what is the reference for this lapse rate?

**We report now that this lapse rate represents the mean atmospheric lapse rate, for example, referenced by Kaser at al. (2010).**

**Comment 20**

- 426: many studies on rain-on-snow events set a minimum snow-depth threshold to define a rain event as a rain-on-snow event, especially because snow tends to be patchy when it is shallow. Could you comment on this in the manuscript?

**It is true that rain-on-snow events are defined by setting a snow depth threshold (for example, above 25 cm (Würzer et al., 2016) and a minimum amount of liquid precipitation falling during a specific period of time, normally 24 hours. In this context, however, we simply wanted to describe that rain was falling on a snowpack. Therefore, we removed the term "rain-on-snow" and replaced it by "precipitation" event.**

**Comment 21**

- 431: again, replace "was more variable" and "slightly increased" with some quantitative statements.

**We modified as follows: "Also turbidity slightly increased from 4.1 to 8.3 NTU during both days".**

**Comment 22**

- 451: how did you quantitatively conclude that the EC-discharge relationship was "the strongest"?

**We decided to remove this sentence and rephrase as follows: "Finally, we evaluated the hysteretic pattern of discharge and EC in more detail by comparing it against $T_{max}$, $G_{max}$ and the snow presence"**

**Comment 23**

- 484: replace "probably needs to be excluded" with "was excluded" if results support this.

**We think that the expression is more suitable in this context. Therefore, we did not change it.**

**Comment 24**

- 493: replace this with the actual concentrations. Same at line 494 (how much more enriched?)

We modified as follows: "Although these samples did not contain high concentrations of Cd, Ni, and Pb (average concentration: 24.5, 10.2, and 9.6 µS cm-1, respectively), snowmelt in contact with the soil surface was more enriched in such elements (150, 191, and 15 µS cm-1, respectively) than dripping snowmelt."

**Comment 25**

- 520: replace "it is more likely" with something more informative (or just remove it)

**We removed it.**

**Comment 26**

- 565 – 566: could some of these factors be addressed just based on a DEM and some assumptions on radiation distribution, as often done in hydrology to distribute radiation across the landscape?

**We agree that topography-based indices about radiation distribution could be utilized, but their use in the present paper would not fit its current scope, and would extend it too much in our opinion.**

**Comment 27**

- 585 – 588: could you be more specific here with regard to how "Tracer dynamics of EC and stable isotopes associated with monthly discharge variations generally followed the conceptual model of the seasonal evolution of streamflow contributions, as described for catchments with a glacierized area of 17 % (Penna et al. 2017) and 30 % (Schmieder et al. 2017"? in other words, could you replace this interpretative statement with some quantitative results that could allow one to understand "how" and "how much" observed dynamics followed the conceptual models? Also, could you quantify what you mean with "isotopic dynamics were generally less pronounced compared to these studies"?

**We added the following sentence to make the statement clearer: "([]for example, isotopic depletion and low EC during snowmelt period in June, less isotopic depletion and low EC during glacier melt period)[]". However, we think that quantitative results cannot be given here as the hydrochemical dynamics of those three catchments (our study and the two references) are very specific with respect to temporal and spatial variability (all these studies were carried out in different years). For this reason, we describe that the "Tracer dynamics of EC and stable isotopes [] generally followed the conceptual models".**

**Related to the last aspect, we removes "generally" and added that the tracer dynamics vary also among different sampling years. This is underlined by the previous reply.**

**Comment 28**

- 646-648: this sentence seems recursive to me

**This sentence seems to repeat partly the previous one but we think that first reporting the general agreement of the conceptual models and then stating the constraint of this agreement is due to the glacierized extent and catchment size is important.**

**Comment 29**

 - 682ff: I think no field work will be ever able to capture all potential variability of hydrologic processes. If the representativeness of this campaign is something that should be discussed, that this should be done in greater details and probably earlier in the manuscript.

**We fully agree on this comment and hope that our work will contribute to narrow this lack of research. The representativeness is always an important point that needs to be addressed. In this context, we want to underline that the sampling schemes was designed to respect the spatial variability of hydrochemistry. Therefore, we added a sentence regarding the representativeness with respect to the sampling scheme in section 2.3: "
[revised manuscript text omitted]